# 14-3-3 proteins activate *Pseudomonas* exotoxins-S and -T by chaperoning a hydrophobic surface

Tobias Karlberg[1], Peter Hornyak[1], Ana Filipa Pinto[1], Stefina Milanova[2], Mahsa Ebrahimi[1], Mikael Lindberg[3], Nikolai Püllen[1], Axel Nordström [1], Elinor Löverli[1], Rémi Caraballo[4], Emily V. Wong[5,6], Katja Näreoja[1], Ann-Gerd Thorsell[1], Mikael Elofsson[4], Enrique M. De La Cruz[5], Camilla Björkegren [1,2] & Herwig Schüler [1]

*Pseudomonas* are a common cause of hospital-acquired infections that may be lethal. ADP-ribosyltransferase activities of *Pseudomonas* exotoxin-S and -T depend on 14-3-3 proteins inside the host cell. By binding in the 14-3-3 phosphopeptide binding groove, an amphipathic C-terminal helix of ExoS and ExoT has been thought to be crucial for their activation. However, crystal structures of the 14-3-3β:ExoS and -ExoT complexes presented here reveal an extensive hydrophobic interface that is sufficient for complex formation and toxin activation. We show that C-terminally truncated ExoS ADP-ribosyltransferase domain lacking the amphipathic binding motif is active when co-expressed with 14-3-3. Moreover, swapping the amphipathic C-terminus with a fragment from *Vibrio* Vis toxin creates a 14-3-3 independent toxin that ADP-ribosylates known ExoS targets. Finally, we show that 14-3-3 stabilizes ExoS against thermal aggregation. Together, this indicates that 14-3-3 proteins activate exotoxin ADP-ribosyltransferase domains by chaperoning their hydrophobic surfaces independently of the amphipathic C-terminal segment.

[1] Department of Biosciences and Nutrition, Karolinska Institutet, Hälsovägen 4c, 14157 Huddinge, Sweden. [2] Department of Cellular and Molecular Biology, Karolinska Institutet, Berzelius väg 35, 17165 Solna, Sweden. [3] Protein Expertise Platform, Umeå University, Kemihuset, 90187 Umeå, Sweden. [4] Department of Chemistry, Umeå University, Kemihuset, 90187 Umeå, Sweden. [5] Molecular Biophysics and Biochemistry, Yale University, New Haven, CT 06520, USA. [6] Present address: University of California, San Francisco Medical School, Department of Biochemistry and Biophysics, San Francisco, CA 94158, USA. Correspondence and requests for materials should be addressed to H.S. (email: herwig.schuler@ki.se)

**P**seudomonas aeruginosa is an opportunistic pathogen that is infamous for causing hospital-acquired airway and wound infections. To initiate the infection process, the bacterium uses a type III secretion system to deliver a small set of exotoxins into the host cell.[1,2] Two of these, exotoxins-S and -T (ExoS; ExoT), are homologous enzymes consisting of an N-terminal GTPase-activating protein (GAP) domain (73% identity; 81% similarity) and a C-terminal ADP-ribosyltransferase (ART) domain (78% identity; 87% similarity).[3,4] Their GAP domains target Rho-family GTPases which leads to a remodeling of the host actin cytoskeleton. The activities of the ART domains are directed toward a more diverse set of proteins. ExoS targets Ras- and Rho-family GTPases,[5,6] ezrin/radixin/moesin (ERM) proteins,[7] and the intermediate filament protein, vimentin.[8] ADP-ribosylation disturbs these targets presumably by placing the bulky ADP-ribose moiety in a protein–protein interaction (PPI) site, and has multiple consequences including disruption of actin polymers and their anchorage with focal adhesions leading to cell rounding. The cytotoxic effects of ExoS activity are likely due to disruption of Ras effector pathways.[9] ExoT targets the adaptor molecules, CT10 Regulator of Kinase (CRK) and CRK-like (CRKL) and its activity leads to severe changes in cell morphology, presumably by rearrangement of the actin cytoskeleton.[10] Most eukaryotic cell types utilize intrinsic ADP-ribosylation to regulate various vital processes, but exotoxin targets are not among the targets identified for human ADP-ribosyltransferases, and human ADP-ribosyl glycohydrolases appear to be unable to remove exotoxin-mediated ADP-ribosylation.[11] Inhibitors of ExoS ADP-ribosyltransferase activity may be viable for therapeutic intervention, as the ExoS ART domain with an intact catalytic activity is needed for bacterial dissemination in the Pseudomonas-infected lung.[12]

The ADP-ribosyltransferase activity of ExoS and ExoT and their cellular toxicity are dependent upon association of the ART domains with 14-3-3 proteins.[13,14] These are ubiquitous, abundant regulators of diverse signaling pathways, and act through hundreds of cellular targets. Seven human genes encode 14-3-3 isoforms, and post-translational modification and both homo- and heterodimerization contribute to the complexity of 14-3-3 functions.[15–17] 14-3-3 proteins have extended hydrophobic surface patches[18] and have recently been recognized to have chaperone-like functions.[15] However, their bona fide regulatory functions are mediated by recognition of phosphoserine or phosphothreonine residues in their targets. A number of crystal complex structures illustrate how target protein-derived phosphorylated peptides bind to an amphipathic groove inside the bowl-shaped 14-3-3 dimer.[17] By contrast, the interaction of ExoS with 14-3-3 proteins is phosphorylation independent, although it is mediated by a C-terminal segment of ExoS binding in the same groove.[19–21] Previous work has clarified the importance of the amphipathic side chains in a short sequence motif, [426]LDLA[429] in ExoS, in the interaction with 14-3-3 proteins and in Pseudomonas virulence.[22–24] This motif (hereafter designated LDLA-box) is conserved in ExoT as well as in the putative 14-3-3 activated bacterial orthologues, AexT[25] and VopT.[26] The extended LDLA-box has also been used as a template for synthetic ExoS inhibitory peptides.[27,28] But whether and how 14-3-3 binding to this short amphipathic sequence is connected to the catalytic activity of the toxins is unknown.

Here we present crystal structures of human 14-3-3β in complex with the ART domain of both ExoS and ExoT. This identifies an extensive hydrophobic-binding interface. ExoS constructs terminating at S419, and thus lacking the LDLA-box, binds 14-3-3β with sub-micromolar affinity; they are inactive when the proteins are reconstituted, but have bona fide ExoS activity when co-expressed with 14-3-3β. We conclude that the core of the ExoS ART domain is competent for catalysis. This is supported by swapping the ExoS C-terminal segment including the LDLA-box with the C-terminal fragment of the 14-3-3 independent toxin Vis from Vibrio splendidus. The resulting chimeric toxin has ADP-ribosyltransferase activity toward ExoS targets in the absence of 14-3-3. Recombinant ExoS is prone to aggregation, and 14-3-3 stabilizes ExoS during thermal aggregation. Collectively, our results indicate that 14-3-3 proteins act as exotoxin chaperones rather than activators.

## Results

**Structures of the ExoS:14-3-3β and ExoT:14-3-3β complexes.** ADP-ribosyltransferase activity of ExoS and ExoT requires 14-3-3 binding, but the mechanism of activation was unknown, and no structural models were available for any 14-3-3 activated toxin. We addressed these problems by X-ray crystallography and biochemical analyses. To identify suitable combinations of exotoxin ART domains and 14-3-3 isoforms to study, we evaluated 14-3-3 concentration-dependent exotoxin activities in vitro. Six of the seven human 14-3-3 isoforms stimulated ExoS enzymatic activity to similar extent; 14-3-3σ had markedly lower apparent affinity, and 14-3-3β was the isoform with the highest apparent affinity for the toxin (Supplementary Fig. 1). We co-expressed 14-3-3β with either ExoS$^{E379A,E381A}$ or ExoT$^{wt}$ ART domains in Escherichia coli (Fig. 1a; for details, see the Supplementary Methods section). Using immobilized metal ion chromatography we isolated hexahistidine-tagged ADP-ribosyltransferase domains of either toxin—ExoS$^{233-453(E379A,E381A)}$ and ExoT$^{235-435(wt)}$—in complex with untagged 14-3-3β. For both exotoxins, ion exchange chromatography (IEC) on heparin sepharose, as well as consecutive size exclusion chromatography (SEC) coupled to right-angle light scattering (SEC-RALS), indicated the formation of protein complexes of various constitutions. A protein complex with an apparent molecular weight of roughly 83.5 kDa (indicative of a heterotrimer: an ExoS$^{E379A,E381A}$ monomer bound to a 14-3-3β dimer; expected molecular weight 83246 Da; Fig. 1b) readily crystallized under various conditions. We obtained crystals in space group C2 that diffracted to 3.22 Å and contained one heterotrimer in the asymmetric unit. We solved the structure using molecular replacement with 14-3-3β and Vis toxin (PDB entries 2C23[21] and 4XZJ[29]) as search models (Supplementary Table 1). A complex with an apparent molecular weight of roughly 95 kDa (heterotetramer; expected molecular weight 104206 Da; Fig. 1d and Table 1) yielded crystals of space group P2$_1$2$_1$2 that diffracted to 3.24 Å (Fig. 1e). Subsequent work (see Supplementary Methods for details) resulted in crystal structures of the 14-3-3β:ExoS apo heterotrimer at 2.34 Å (Fig. 1c), the heterotrimer with the non-hydrolyzable NAD$^+$ analog, carba-NAD in the active site at 2.50 Å (Fig. 1f, g), and the heterotrimer with inhibitor STO1101 (ref. [30]) at 3.24 Å resolution (Supplementary Fig. 2). The 14-3-3β:ExoT complexes showed a similar distribution of various molecular weight species (Supplementary Figs. 3, 4). We determined the structure of the 14-3-3β:ExoT heterotetramer in complex with STO1101 (Fig. 1h); however, the model could not be refined beyond ~3.80 Å resolution. The 14-3-3β complexes of ExoS and ExoT looked very similar overall (Supplementary Fig. 5), and the STO1101-bound ART domains alone (C chains) aligned with a root mean square deviation of 0.97 Å.[31]

**A hydrophobic interface between 14-3-3 and the ART exotoxins.** Our crystal structures revealed an extensive interface between 14-3-3 and the ExoS and ExoT ART domains. This site

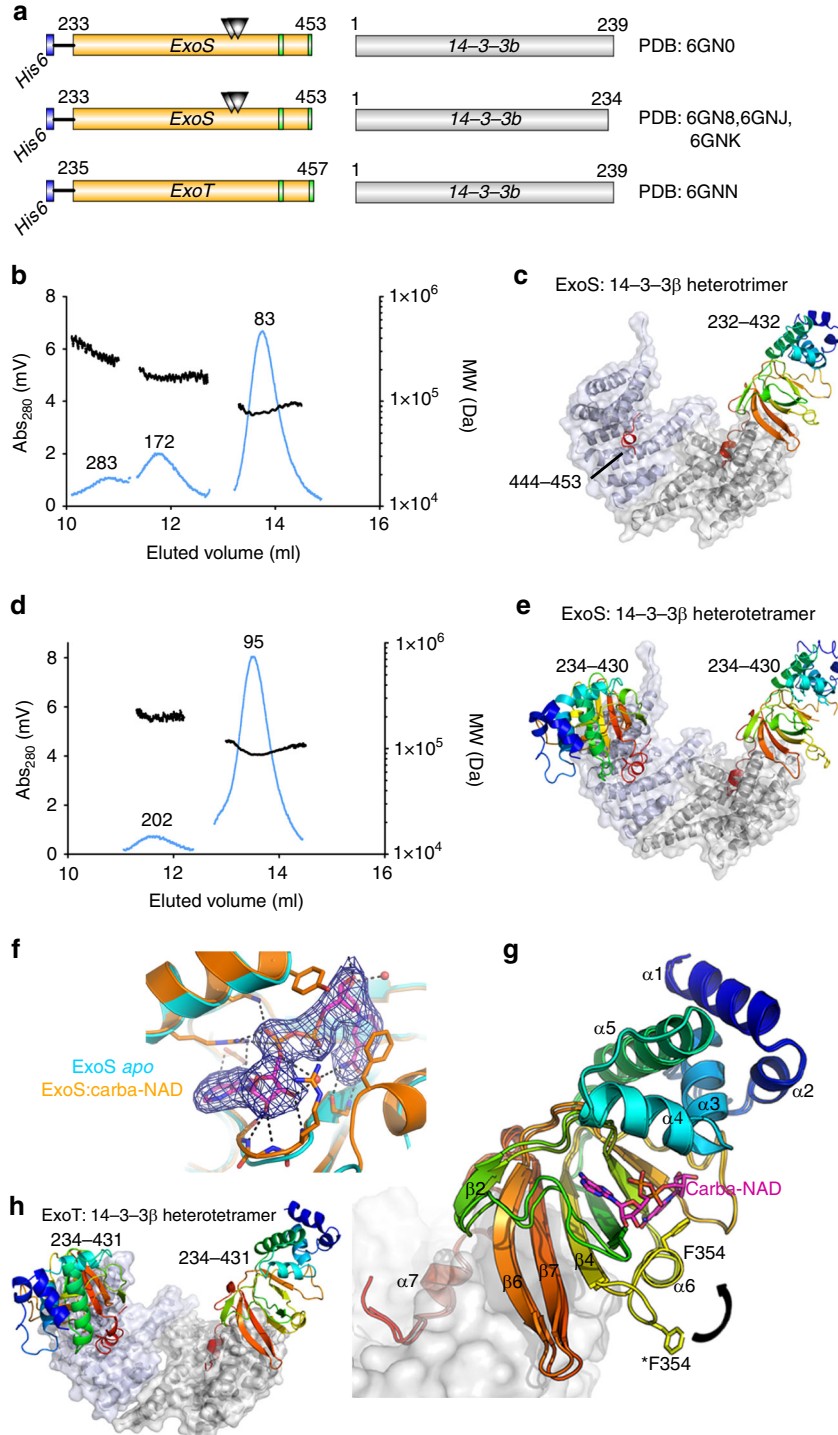

**Fig. 1** Crystal structures of 14-3-3β:ExoS and 14-3-3β:ExoT complexes. **a** Schematic representation of co-expression constructs used for structure determination. Triangles represent the E379A,E381A double mutation. **b, d** SEC-RALS profiles of the heterotrimeric (**b**) and heterotetrameric (**d**) 14-3-3β: ExoS complexes (main peaks). Estimated molecular weights are indicated. **c, e** Structures of the 14-3-3β:ExoS heterotrimer (**c**) and heterotetramer (**e**). The 14-3-3β dimer is shown in gray. Numbers in italics indicate the ExoS residues that were resolved in the electron density. **f** Detail of the electron density around carba-NAD in the active site of ExoS ($2F_{obs} - F_{calc}$ electron density map contoured at $1\sigma$ ($0.1073 \, e\text{Å}^{-3}$)). **g** Comparison of the ExoS apo and carba-NAD structures. The loop containing F354 is in two different conformations, where the open conformation from the apo structure is marked with an asterisk. Secondary structural elements are numbered. **h** Structure of the 14-3-3β:ExoT heterotetramer. **c, e, g, h** ExoS respectively ExoT are shown colored in a gradient from the N-terminus (blue) to the C-terminus (red) of the construct

**Table 1 Data collection and refinement statistics**

|  | ExoS E379A,E381A apo:14-3-3β heterotetramer (PDB 6GN0) | ExoS E379A,E381A apo:14-3-3β heterotrimer (PDB 6GN8) | ExoS E379A,E381A -STO1101:14-3-3β heterotrimer (PDB 6GNJ) | ExoS E379A,E381A -carba-NAD:14-3-3β heterotrimer (PDB 6GNK) | ExoT-STO1101:14-3-3β heterotetramer (PDB 6GNN) |
|---|---|---|---|---|---|
| **Data collection**[a] |  |  |  |  |  |
| Beamline | ESRF, ID30A3 | Diamond, i03 | Diamond, i03 | Diamond, i03 | Diamond, i03 |
| Wavelength (Å) | 0.96770 | 0.97624 | 0.97624 | 0.91841 | 0.97625 |
| Space group | $P2_12_12$ | C2 | C2 | C2 | $P2_12_12$ |
| Unit cell dimensions (Å, Å,Å,°,°,°) | 165.20, 168.76, 82.64, 90, 90, 90 | 159.91, 59.40, 120.33, 90, 125.79, 90 | 160.40, 59.36, 120.64, 90, 125.84, 90 | 160.98, 56.78, 120.40, 90, 126.46, 90 | 115.38, 60.30, 81.19, 90, 90, 90 |
| Resolution (Å) | 118.1–3.24 (3.29–3.24) | 97.6–2.34 (2.38–2.34) | 50.0–3.24 (3.43–3.24) | 48.4–2.50 (2.65–2.50) | 57.7–3.79 (3.85–3.79) |
| Unique reflections | 37506 (1875) | 39047 (1928) | 14937 (2326) | 30578 (4821) | 6026 (351) |
| R (merge) | 0.510 (2.40) | 0.092 (1.39) | 0.351 (1.73) | 0.070 (1.28) | 0.205 (2.30) |
| Completeness (%) | 99.8 (96.7) | 99.9 (98.9) | 98.7 (95.6) | 99.2 (97.8) | 100 (99.0) |
| Redundancy | 13.3 (14.0) | 6.5 (5.5) | 6.8 (6.7) | 6.0 (5.7) | 12.0 (12.2) |
| $<I>/<\sigma I>$ | 5.0 (1.4) | 10.4 (1.2) | 5.1 (1.0) | 14.2 (1.1) | 7.3 (1.2) |
| CC(1/2) | 0.980 (0.528) | 0.999 (0.621) | 0.988 (0.468) | 0.999 (0.710) | 0.998 (0.515) |
| **Refinement** |  |  |  |  |  |
| Resolution (Å) | 118.1–3.24 (3.33 -3.24) | 97.6–2.34 (2.40-2.34) | 48.9–3.24 (3.50–3.24) | 48.4–2.50 (2.59–2.50) | 53.4–3.79 (4.24-3.79) |
| R-factor (%) | 24.35 | 19.95 | 17.22 | 20.23 | 28.52 |
| Reflections used for R-factor | 35313 | 37003 | 13772 | 29047 | 5621 |
| R-free (%) | 29.44 | 24.77 | 22.17 | 22.76 | 38.49 |
| Reflections used for R-free | 1829 | 1957 | 1037 | 1531 | 314 |
| R.m.s.d. bond length (Å)[b] | 0.010 | 0.010 | 0.010 | 0.010 | 0.003 |
| R.m.s.d. bond angle (°)[b] | 1.2 | 1.1 | 1.1 | 1.1 | 0.6 |
| Wilson B-factor (Å²) | 52.61 | 54.93 | 128.40 | 92.00 | 92.59 |
| Mean B-factor (Å²) | 54.42 | 73.18 | 102.91 | 119.02 | 128.08 |
| **Ramachandran plot**[c] |  |  |  |  |  |
| Most favored (%) | 90.9 | 96.2 | 94.7 | 97.4 | 85.1 |
| Disallowed (%) | 1.6 | 0.6 | 0.6 | 0.3 | 3.4 |

[a]All data sets collected from single crystals
[b]Using the parameters of Engh and Huber[54]
[c]From MolProbity[55]

involves the C-terminal α-helices-8 and -9 of 14-3-3β, and can be sub-divided into two parts: (i) The two proteins bury a common hydrophobic core made up of side chains originating from the 14-3-3β C-terminal helices and the central β-sheet of ExoS/T (primarily strands β4, β7, and β8; Fig. 2a–c). This sub-site constitutes roughly half of the total contact area between the two proteins (ExoS: 762 Å² out of 1580 Å²; ExoT: 766 Å² out of 1582 Å²).[32] (ii) The short segment that connects the LDLA-box containing exotoxin helix α7 with the core of the ART domain forms an extensive network of hydrogen bonds with the turn between 14-3-3β helices α8 and α9. The entire interface between 14-3-3β and the ART domain is similar in ExoS and ExoT (Supplementary Fig. 5) and, based on protein sequence comparison, appears to be well conserved in other 14-3-3-dependent toxins (Supplementary Fig. 6).

This hydrophobic interface between the exotoxins and 14-3-3 proteins might be a useful target for development of ExoS and ExoT inhibitors. Fortuitously, one of only few potential PPI inhibitors known to bind 14-3-3 outside of the amphipathic groove, the compound NV1, was shown to bind at the rim of the hydrophobic ExoS/T-binding site (Fig. 2c).[33] We found that STO1704, a compound related to NV1, inhibited K-Ras modification by ExoS[233–453] and CRK modification by ExoT[235-457] in a dose-dependent manner (Fig. 2d and Supplementary Fig. 7). The potency is low (IC$_{50}$ 638 and 726 μM, respectively); nevertheless, this provides proof-of-principle in support of PPI inhibitors as ExoS/T inhibitors.

Our crystal structures of the 14-3-3β:ExoS and the 14-3-3β: ExoT ART domain complexes all revealed the bowl-shaped 14-3-3 dimer that aligned well with many previous structures of 14-3-3 isoforms (root mean square deviations of less than 1 Å over approximately 230 Cα-pairs). Neither the 14-3-3β dimer nor the toxin-bound 14-3-3β protomer show any significant conformational differences compared to the previous ExoS peptide bound

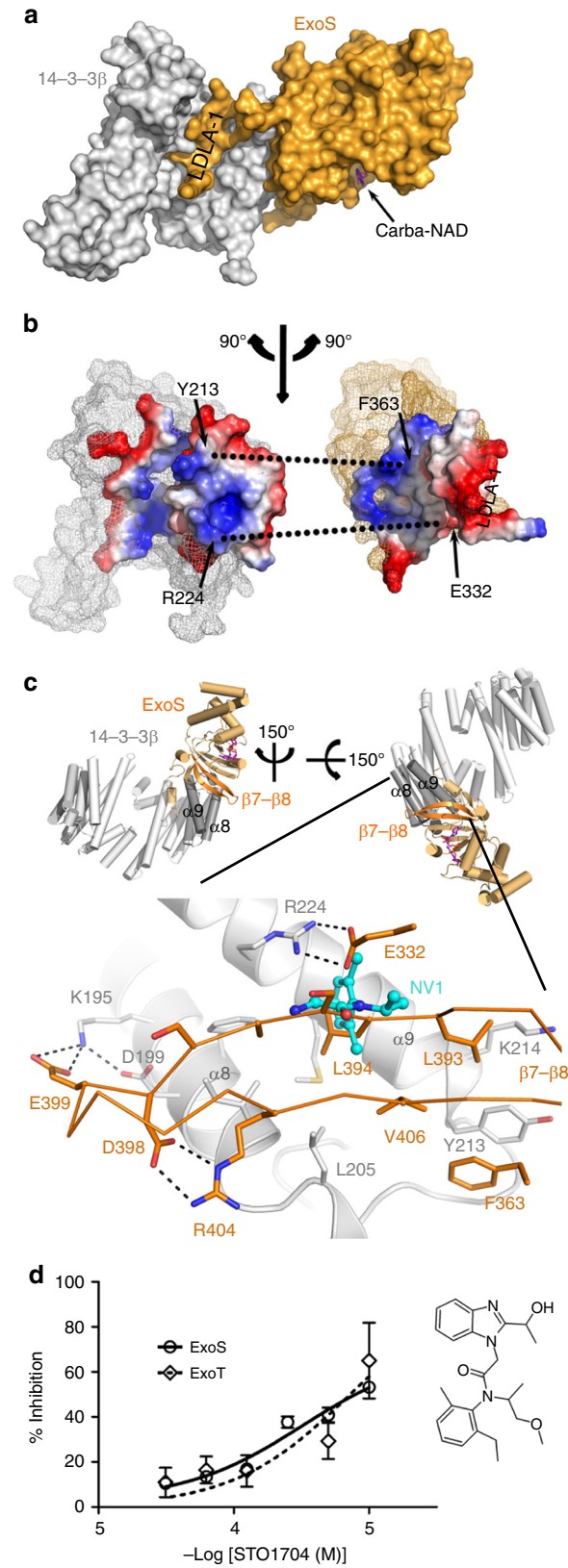

**Fig. 2** A hydrophobic interface between 14-3-3 and exotoxins-S and -T. **a** Surface representation of the ExoS:14-3-3β complex. ExoS is shown in gold and one monomer of the 14-3-3β homodimer is shown in gray. The position of LDLA-box 1 in the amphipathic groove is indicated. **b** The partners in the complex rotated 90° away from each other to display the binding interface, colored to show surface charge distribution (blue, positive charges; red, negative charges; white, non-polar). Two key interactions are indicated to facilitate orientation. **c** Detail of the ExoS:14-3-3β interaction around ExoS β7–β8. The inset on top indicates the orientation of the main panel: The complex (as in Fig. 1c) was rotated to the left and then down toward the viewer. Main panel: Side chains of each protein that participate in interactions are shown (gold, ExoS; gray, 14-3-3β) and hydrogen bonds are indicated as dots. The binding site for PPI inhibitor NV1 (ref. [33]) is indicated by the NV1 structure (carbons in cyan). **d** Concentration-dependent inhibition of ExoS K-Ras modification, and ExoT Crk modification, by the NV1 analog, STO1704 (means ± s.e.m.; $n = 3$). The chemical structure of STO1704 is shown

structure: The ART domain and its C-terminal extension appear to wrap onto the solid scaffold of the 14-3-3 dimer.

**A second LDLA-box binds 14-3-3 in the heterotrimeric complex.** In all our protein complex structures, the phosphoprotein-binding site of the exotoxin bound 14-3-3β protomer was occupied by an amphipathic peptide containing the LDLA-box in a short α-helical segment (ExoS residues $^{420}$QGLLDALDLAS$^{430}$; Fig. 3a) in which the D427 side chain hydrogen bonded with the 14-3-3β Y127 and Y130 side chains, and the hydrophobic side chains formed hydrophobic interactions within the amphipathic groove of 14-3-3β (Fig. 3b). Both positioning and interaction pattern were virtually identical in previous crystal structures of ExoS-derived peptides in complex with 14-3-3 proteins.[21,23]

ExoS, ExoT, and AexT of the 14-3-3-dependent ADP-ribosylating toxins contain a second LDLA motif (ExoS residues 450–453; designated LDLA-box 2) in their very C-termini (Fig. 3a and Supplementary Fig. 6). In one of our crystal structures of the 14-3-3β:ExoS heterotrimer, the residues connecting LDLA-box-1 and -2, E433-D442, could not be modeled; but we found unambiguous electron density placing LDLA-box 2 in the phosphoprotein-binding site of the second 14-3-3β protomer (Figs. 1c, 3c). LDLA-box 2 bound in a configuration that was highly similar to the configuration of LDLA-box 1 in all four structures. This finding prompted us to examine the contribution of LDLA-box 2 to 14-3-3 binding. We used fluorescence anisotropy to estimate the affinities of 14-3-3β to N-terminal green fluorescent protein (GFP) fusions of ExoS$^{233-435}$ (containing only LDLA-box 1) and ExoS$^{233-453}$ (containing both LDLA-box-1 and -2, and the native C-terminus). The results indicate comparable affinities in the nanomolar range (~40 nM for GFP-ExoS$^{233-453}$; ~70 nM for GFP-ExoS$^{233-435}$; Fig. 3d and Supplementary Fig. 8). Similar estimates ($K_{d,app} = 22.6$, respectively, 34.7 nM for the β and ζ isoforms) were derived from the 14-3-3 concentration-dependent activation of ExoS activity (Fig. 3e and Supplementary Fig. 1).

These results were substantiated by in vivo toxicity assays using *S. cerevisiae*. The yeast has two orthologs of 14-3-3 proteins, Bmh1 and Bmh2, and the side chains contributing to the ExoS interactions in our crystal structures are largely conserved in both. Galactose-induced expression of either ExoS$^{233-435}$ or ExoS$^{233-453}$ conferred toxicity such that no colonies were formed on galactose-containing agar plates (Fig. 3f). Conversely, expression of ExoS$^{233-419}$ (lacking both LDLA-boxes) had no effect on

structures.[21,23] Thus our structures show that the LDLA-box in the context of the ART domain binds to 14-3-3 as the LDLA-box containing peptides, and in binding, does not distort the 14-3-3

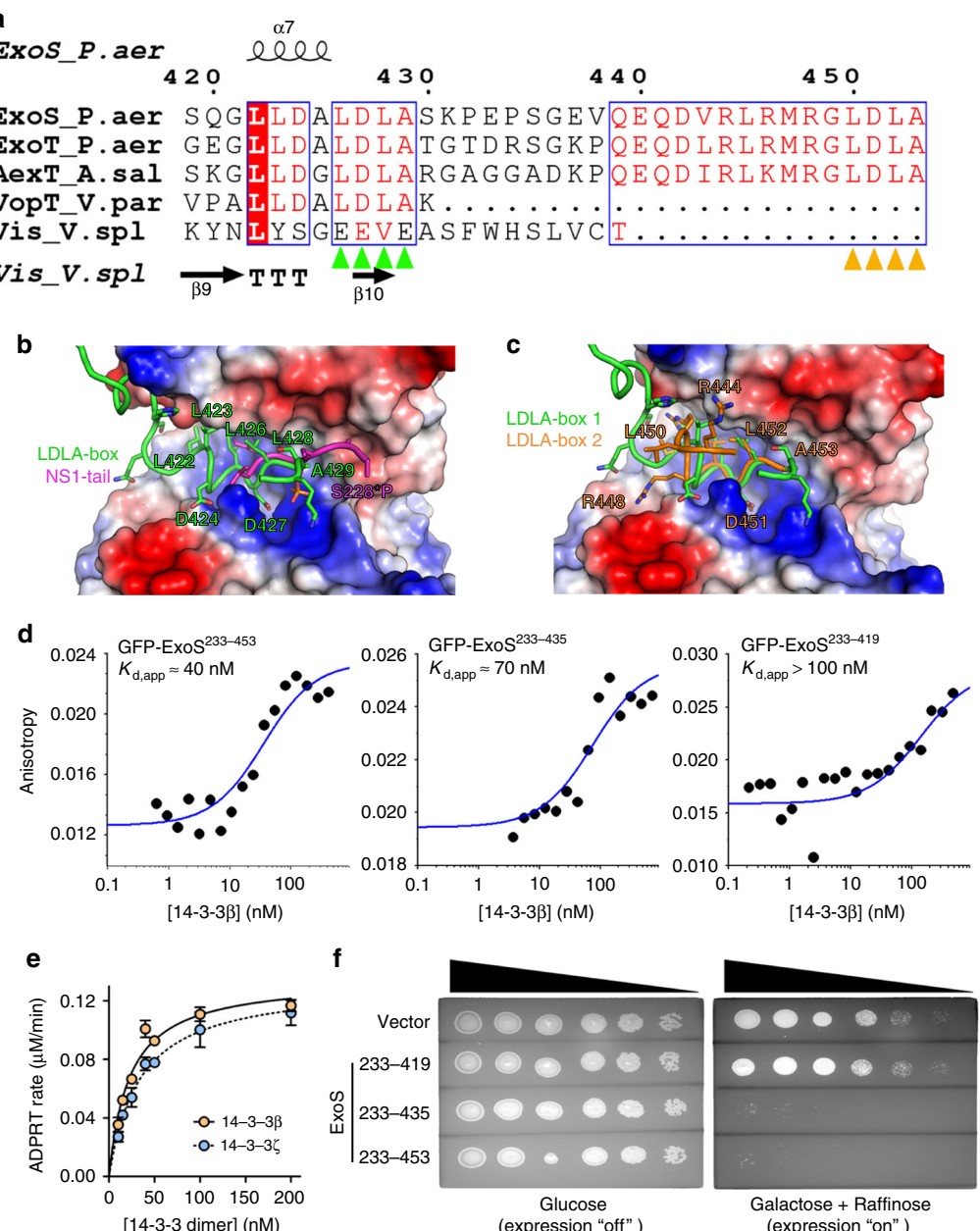

**Fig. 3** A second LDLA-box is capable of binding the second 14-3-3 protomer. **a** Sequence alignment of 14-3-3-dependent toxins ExoS, ExoT, AexT, and VopT and, for comparison, the 14-3-3 independent Vis toxin. Presence of LDLA-box 1 (green arrowheads) unites all 14-3-3-dependent toxins; several also feature a second LDLA-box (orange arrowheads) at their very C-terminus. **b** ExoS LDLA-box 1 binding to the amphipathic groove of 14-3-3β and comparison to phosphopeptide binding at the same site (PDB entry 4O46). **c** ExoS LDLA-box-1 and -2 binding to the two 14-3-3β protomers. **d** Anisotropy of GFP-tagged ExoS constructs indicated tight binding of the ExoS ART domain (residues 233–453) to 14-3-3β. Truncation of LDLA-box 2 (residues 233–435) led to a slight reduction in apparent affinity. Truncation of both LDLA-boxes resulted in a protein (residues 233–419) that bound 14-3-3β with sub-micromolar affinity using only the hydrophobic binding site (Fig. 2). **e** 14-3-3 concentration-dependent activation of ExoS$^{233-453}$ activity (means ± s.e. m.; $n = 2$), confirming the sub-micromolar affinity measured by fluorescence anisotropy (**d**). See Supplementary Fig. 1 for further details. **f** Yeast toxicity assay of ExoS. Five-fold serial dilutions of yeast cells spotted on agar containing either glucose (target gene expression repressed) or galactose/raffinose (expression induced). Expression of the ExoS ADP-ribosyltransferase domain is highly toxic, unless the C-terminal segment ExoS$^{420-453}$, containing both LDLA-boxes, is absent

colony formation compared to presence of the empty vector. We concluded that LDLA-box 2 binding to the second 14-3-3 protomer was an intriguing observation with possible implications for ExoS regulation. It is conceivable that LDLA-box 2 might participate in the formation of higher molecular weight 14-3-3:exotoxin complexes (Supplementary Fig. 3), or fold onto the ART domain. If such an event was coupled to catalytic activity, it could provide a layer of regulation; possibly, multimerization acts as a sensor for the presence of high amounts of exotoxin, i.e.

conditions under which heterotetramer formation would liberate LDLA-box 2 from its binding site.

**Mechanism of toxin activation.** The importance of LDLA-box 1 binding to the amphipathic groove of 14-3-3 is undisputed,[22–24] but its role has been unknown. Given the general sequence homology between these exotoxins and ADP-ribosylating toxins of known structure, we initially expected that 14-3-3 binding might unlock an active exotoxin conformation by sequestering

the LDLA-box containing C-terminus. However, we could not reconcile the importance of LDLA-box 1 with our crystal structures. Site-directed mutagenesis did not support a role for a structural link between 14-3-3 and the NAD$^+$-binding site in regulation of catalytic activity (Supplementary Fig. 9 and Supplementary Table 2). Therefore, we asked whether the putative activating role of 14-3-3 could be substituted by the short carboxy-terminal sequence of a 14-3-3 independent but otherwise homologous toxin. Based on available crystal structures, we designed a chimeric toxin that comprised ExoS residues 233–409, followed by the C-terminus (residues 218–249)[34] of Vis toxin (Fig. 4a). This construct, designated SxVis, displayed NAD$^+$ glycohydrolase activity, and ADP-ribosylated K-Ras in the absence of 14-3-3 (Fig. 4b). Neither of these activities could be further stimulated by 14-3-3. ExoS alone had no activity without 14-3-3 present (Fig. 4c). Although SxVis catalytic efficiency ($k_{cat}$/$K_M$) of K-Ras modification was >40-fold lower than that of 14-3-3 activated ExoS, SxVis had a $K_M^{NAD}$ in the same range as ExoS (Fig. 4c and Table 2) indicating a fully functional active site. These kinetics also indicate that LDLA-box 1 is involved in either target protein recognition or catalysis, but not NAD$^+$ binding.

These findings, in combination with the observation (Fig. 3d) that 14-3-3β had considerable affinity for ExoS$^{233–419}$ (lacking both LDLA-boxes; Fig. 4d), prompted us to co-express ExoS$^{233–419}$ with 14-3-3β. We found that expression of this protein complex elicited toxicity in *E. coli*, and this toxicity was dependent on the presence of the catalytic glutamates (Fig. 4e). This was in line with our initial observation that expression of complex containing the full wild-type ExoS ART domain was toxic (Supplementary Methods). However, we could identify conditions under which protein complex could be produced (see Methods). We found that the native complex of ExoS$^{233–419}$ and 14-3-3β, when purified from cells co-expressing the proteins, ADP-ribosylated K-Ras and had NAD$^+$ glycohydrolase activity, whereas ExoS$^{233–419}$ when reconstituted with 14-3-3β was inactive (Fig. 4f–h and Supplementary Figs. 10–13). These findings suggested that LDLA-box 1 is dispensable for ExoS activity, but is needed for other reasons. Since bacterial expression of all exotoxin constructs induced the formation of inclusion bodies, we studied the effect of 14-3-3 on ExoS stability. Using dynamic light scattering (DLS), we observed that ExoS$^{233–453}$ formed aggregates, with an onset temperature of 46.7 °C (Supplementary Table 3). Sub-stoichiometric amounts of 14-3-3β were able to elevate the transition temperature of ExoS aggregation (Fig. 4i). The stabilizing effect of 14-3-3β was abrogated by the PPI inhibitor STO1704, whereas carbonic anhydrase, an unrelated protein, had no effect on ExoS aggregation (Supplementary Table 3 and Supplementary Fig. 14). Together, this leads us to conclude that 14-3-3 activation of ExoS does not involve a structural transition to enable substrate or target binding, but rather is a consequence of chaperoning hydrophobic surfaces to prevent the ART domain from aggregating (Fig. j).

## Discussion

While it is well documented that LDLA-box 1 is crucial for exotoxin activity in vivo, we showed here that the ExoS ART domain is intrinsically active, and that LDLA-box 1 is dispensable for 14-3-3-mediated exotoxin activity in vitro. Our results also indicate that the hydrophobic 14-3-3-binding site in the exotoxins is an aggregation prone surface that needs chaperoning. Thus we might assume that in vivo, the exotoxin C-terminus recruits 14-3-3 proteins early during passage through the type 3 secretion system and thereby ensures fast binding of the hydrophobic surface to its chaperone. Some T3SS targets have been shown to

traverse the needle complex starting with the translocation signal containing N-terminus.[35] 14-3-3-dependent exotoxins also contain a translocation signal in their N-termini,[1] but it is unknown whether they enter host cells starting with their N- or their C-terminal end.

Co-expression of ExoS$^{253–419}$ with 14-3-3β was found to be toxic to bacteria (Fig. 4e), while overexpression of the same construct was not toxic to yeast (Fig. 3f). Although the exotoxin interacting residues appear to be conserved in the yeast 14-3-3 orthologs, it is feasible that the short constructs binds these proteins with lower affinity than human 14-3-3 proteins. However, the two experimental systems are sufficiently different to exclude a straightforward conclusion regarding the lack of yeast toxicity of this ExoS construct.

The main contact site between 14-3-3β and the ART domains overlaps with the recently discovered binding interface between the yeast 14-3-3 ortholog, Bmh1, and trehalase.[36] The modes of interaction are different however, as an α-helical segment (S722-G729) of trehalase resides in the hydrophobic groove formed by Bmh1 helices-8 and -9 roughly perpendicular to the orientation of the ExoS/T β-strands in our complexes. Structures of other complexes of 14-3-3 with client proteins have been determined as well,[37–40] but the binding sites for these proteins show less overlap with the ExoS/T-binding site than that for trehalase (Supplementary Fig. 15). In a development of pharmacological interventions involving the hydrophobic exotoxin-binding site, it will be important to address the question of whether the binding of other proteins interacting at this site is affected.

Barbieri and co-workers[41] defined a so-called region A (ExoT$^{246–264}$) near the N-terminus of the domain as critical for substrate recognition by ExoT. Our crystal structures show that regions A of both ExoS and -T fold into two helices (α1 and α2), unlike the corresponding region with low homology in related toxins (Supplementary Fig. 16). We noted differential effects of our F327R mutant depending on whether a proteinaceous substrate or a free arginine analog was available (Supplementary Table 2). As α1 and α2 locate close to the NAD$^+$ containing active site loop, it is plausible that target protein binding at this site would influence the affinity of ExoS for NAD$^+$.

Our crystal structures provide important information for a better understanding of 14-3-3-dependent toxins, and also for the development of enzyme inhibitors as potential anti-infectives. However, our discoveries of an exotoxin chaperone function of 14-3-3, of the hydrophobic interaction site between the proteins, and of the inhibitory effect of STO1704, encourage development of specific PPI inhibitors of 14-3-3-dependent exotoxin activation. This is especially important as compounds binding to an exotoxin-specific site would circumvent the off-target effects expected of inhibitors that bind in the amphipathic groove of 14-3-3 proteins.[42]

## Methods

**Materials**. Fine chemicals and growth media were purchased from SigmaAldrich. 1,$N^6$-fluoresceinyl-NAD$^+$ (fluo-NAD$^+$) was obtained from BioLog, and 1, $N^6$-etheno-AMP from Jena Bioscience. The non-hydrolyzable NAD analog carbanicotinamide adenine dinucleotide (carba-NAD; PubChem ID 112345-60-5) was synthesized as described before,[43] with minor modifications (see Supplementary Methods and Supplementary Fig. 17 for details). $^1$H- and $^{13}$C-NMR spectra (Supplementary Fig. 18) were in agreement with the original report. ExoS/T inhibitor STO1101 (3-(4-oxo-3,5,6,7-tetrahydro-4H-cyclopenta[4,5] thieno[2,3-d]pyrimidin-2-yl)propanoic acid; PubChem ID 412962-43-7) was purchased from Enamine (cat. no. Z96229612).[30] PPI inhibitor STO1704 (N-(2-ethyl-6-methylphenyl)-2-[2-(1-hydroxyethyl)-1H-benzimidazol-1-yl]-N-(2-methoxy-1-methylethyl)acetamide) was purchased from ChemBridge (cat. no. 6944250).

**Molecular cloning**. The cDNA fragments encoding ExoS residues 233–453 and ExoT residues 235–457, sub-cloned in pET28a (Novagen) to obtain an N-terminal

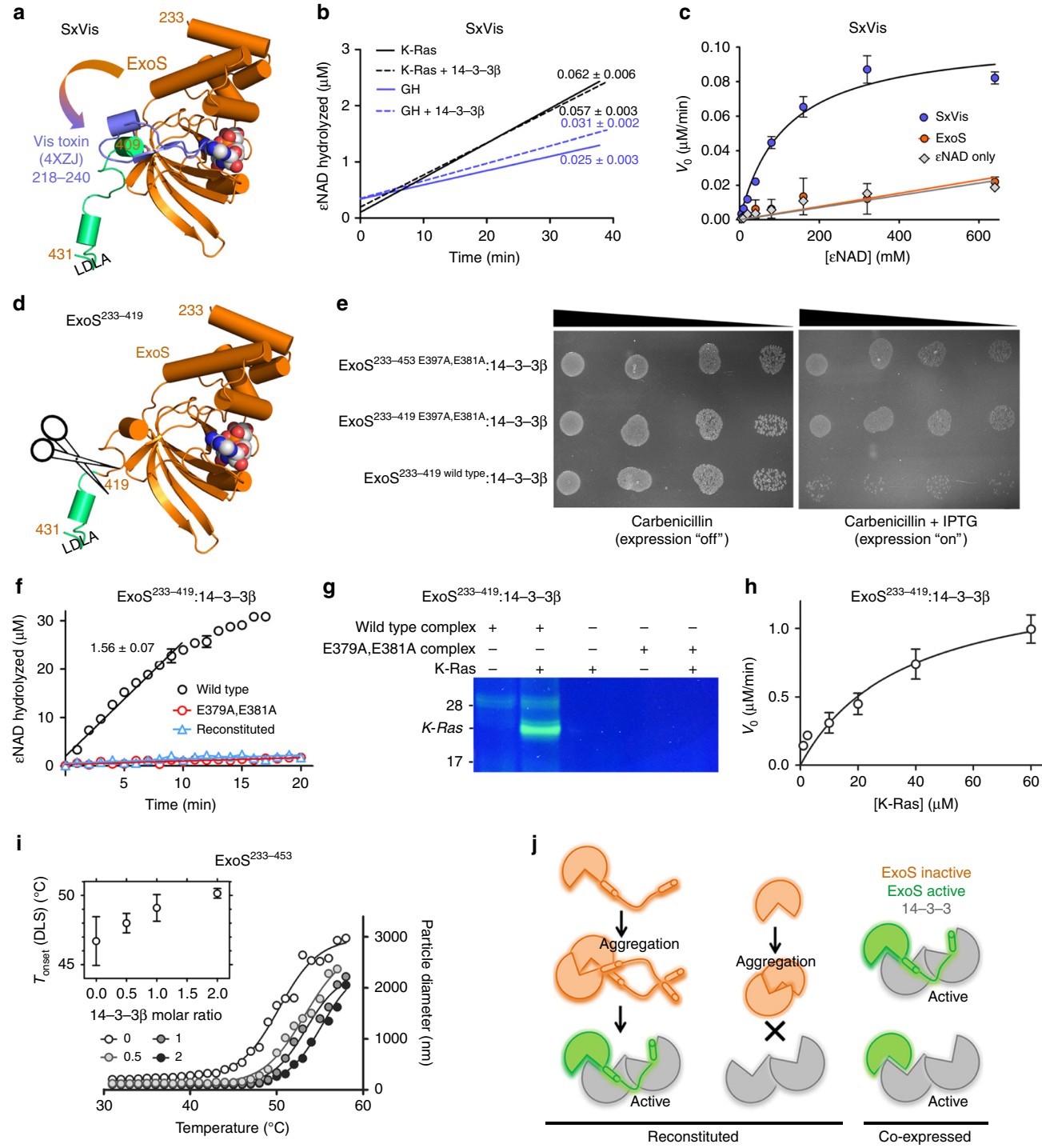

hexahistidine fusion, were contributed by Yngve Östberg (Umeå University). Other exotoxin encoding cDNA fragments were PCR-amplified from extracts of *P. aeruginosa* PAK cells, contributed by Charlotta Sundin and Åke Forsberg (Umeå University). ExoS[419–453] cDNA was sub-cloned in pNic-trx (provided by the Protein Science Facility at Karolinska Institutet/SciLifeLab), encoding an N-terminal thioredoxin followed by a hexahistidine tag; all other exotoxin encoding DNA constructs were sub-cloned in pNic28-Bsa4 (GenBank: EF198106.1) to obtain N-terminal hexahistidine fusion constructs. Expression plasmids containing the 14-3-3 encoding cDNAs YWHAB (pTvHR21-SGC), YWHAG (pTvHR21-SGC), YWHAE (pLic-SGC1), YWHAH (pNic28-Bsa4), YWHAQ (pNic28-Bsa4), YWHAS (pGEX2TK), and YWHAZ (pGEX4T1) were contributed by the Structural Genomics Consortium and obtained from Addgene. The 14-3-3ζ encoding cDNA YWHAZ was also sub-cloned in pNic28-Bsa4 to obtain N-terminal hexahistidine fusion construct. Expression plasmids for other ExoS fragents were based on vector pNIC-Bsa4 or pET28a, and have been described.[30] GFP-tagged

ExoS expression vectors were constructed by sub-cloning ExoS[233–419], ExoS[233–435], ExoS[233–453], and ExoS[419–453] in pET-GFP1a.[44] An ExoS:14-3-3β co-expression vector was constructed by sub-cloning a cDNA fragment encoding the E379A, E381A mutant of ExoS[233–453] and the cDNA encoding 14-3-3β[1–239] into pET-Duet1 (Novagen). A co-expression vector for the wild-type ExoS[233–419] protein was constructed by replacing the ExoS[233–453] cDNA sequence with the wild-type ExoS[233–419] cDNA in pET-Duet1. An ExoT:14-3-3β co-expression vector was constructed by replacing the ExoS cDNA sequence with the wild-type ExoT[235–457] cDNA in pET-Duet1. SxVis (ExoS[233–409] fused to Vis[218–249]) and the ExoS[233–453] mutants were obtained as synthetic clones, sub-cloned in pET151b (GeneArt/ThermoFischer Scientific). The coding regions for human Rnd1 and Rac3 were contributed by Pontus Aspenström (Karolinska Institutet) and Rnd1[1–232] and Rac3[1–192] were sub-cloned in pNic28-Bsa4 to obtain N-terminal hexahistidine fusion constructs. Expression vector pET28-MHL-KRASb was contributed by the Structural Genomics Consortium and obtained from Addgene. Expression vector

**Fig. 4** The LDLA-boxes are dispensable for ExoS activity. **a** Schematic representation of the chimeric protein, SxVis, generated by C-terminal swapping. In the ExoS ART domain (gold), the C-terminal fragment (green) was replaced by that of Vis toxin (blue; PDB entry 4XZJ). The position of the phosphopeptide groove binding LDLA-box 1 is indicated. **b** K-Ras modification (black) and NAD$^+$ glycohydrolase (GH) activity (blue) of SxVis alone (solid lines) and in the presence of 2.5 molar excess of 14-3-3 (dotted lines). Rates (in µM εNAD$^+$ hydrolyzed min$^{-1}$) are indicated. **c** SxVis glycohydrolase activity (no 14-3-3) as a function of εNAD$^+$ concentration. The ExoS ART domain construct ExoS$^{233-453}$ is inactive under these conditions. **d** Schematic representation of ExoS ART domain construct ExoS$^{233-419}$ lacking the C-terminal LDLA-box containing fragment (green). **e** Ten-fold serial dilutions of E. coli cells transformed with ExoS expression vectors, spotted on agar containing either antibiotics (expression vector selection) or antibiotics and isopropyl β-D-1-thiogalactopyranoside (IPTG; expression induced). ExoS$^{233-419}$ elicits toxicity in E. coli when co-expressed with 14-3-3β. **f** K-Ras modification by ExoS$^{233-419}$ co-expressed with 14-3-3β, using εNAD$^+$ as a co-substrate. Rates (in µM εNAD$^+$ hydrolyzed min$^{-1}$) are indicated. Reconstituting ExoS$^{233-419}$ with 14-3-3β yields inactive exotoxin under identical conditions. **g** K-Ras modification by ExoS$^{233-419}$ co-expressed with 14-3-3β, using fluo-NAD$^+$ as a co-substrate. ExoS$^{233-419}$ carrying the E379A,E381A mutation, co-expressed with 14-3-3β, is shown as control. Controls and an uncropped version of the gel are provided as Supplementary Figs. 10–13. **h** Co-expressed ExoS$^{233-419}$:14-3-3β complex activity as a function of K-Ras concentration. Kinetic properties are reported in Table 2. **i** ExoS$^{233-453}$ in the presence of 14-3-3β at a varying molar ratio (0–2, as indicated) was heated at 0.4 K min$^{-1}$ while light scattering was measured using a DLS detector. Inset: Half-temperatures of aggregation as a function of [14-3-3β] added. 14-3-3β stabilizes ExoS$^{233-453}$ against heat induced aggregation. **j** Model of ExoS activation. ExoS has the intrinsic requirements for ADP-ribosyltransferase activity, but is aggregation prone. Aggregation is reversible as long as the LDLA-boxes are accessible to initiate 14-3-3-mediated de-aggregation. Aggregation is prevented by co-expression with 14-3-3, irrespective of the presence of the LDLA-boxes. All error bars represent s.e.m.

**Table 2 Kinetic constants for ExoS and ExoT ADP-ribosyltransferase domain activity**

|  | Target | $K_M^{\varepsilon NAD}$ (µM) | $K_M^{target}$ (µM) | $k_{cat}$ (min$^{-1}$) | $k_{cat}/K_M$ (µM$^{-1}$ s$^{-1}$) |
|---|---|---|---|---|---|
| **ART domains** | | | | | |
| ExoS$^{233-453}$ | —[a] | 28.6 ± 6.6 | n.a.[b] | 9.9 ± 0.72 | 5.7 × 10$^{-3}$ |
| ExoS$^{233-453}$ | K-Ras | 49.0 ± 9.1 | 32.2 ± 9.9 | 7.9 ± 0.5 | 2.7 × 10$^{-3}$ |
| ExoS$^{233-453}$ | Rac3 | 41.7 ± 9.3 | 11.4 ± 4.6 | 11.2 ± 0.8 | 4.5 × 10$^{-3}$ |
| ExoS$^{233-453}$ | Rnd1 | 39.9 ± 18.1 | 4.8 ± 1.9 | 7.9 ± 1.0 | 3.3 × 10$^{-3}$ |
| ExoS$^{233-453}$ | Agmatine | 133 ± 14 | 2760 ± 130 | 53.5 ± 2.2 | 6.7 × 10$^{-3}$ |
| ExoT$^{235-457}$ | Crk | 8.2 ± 1.9 | 16.9 ± 2.6 | 2.1 ± 0.10 | 4.3 × 10$^{-3}$ |
| ExoT$^{235-457}$ | Agmatine | 44.4 ± 5.8 | 660 ± 160 | 0.24 ± 0.01 | 0.1 × 10$^{-3}$ |
| **C-terminal deletions** | | | | | |
| ExoS$^{233-435}$ | —[a] | 61.2 ± 8.9 | n.a. | 7.2 ± 0.4 | 1.9 × 10$^{-3}$ |
| ExoS$^{233-435}$ | K-Ras | 37.2 ± 2.2 | 17.0 ± 3.1 | 11.7 ± 0.22 | 5.3 × 10$^{-3}$ |
| ExoS$^{233-419}$ [c] | —[a] | 88.5 ± 11.1 | n.a. | n.d.[d] | n.d. |
| ExoS$^{233-419}$ [c] | K-Ras | 96.9 ± 18.1 | 36.1 ± 15.7 | n.d. | n.d. |
| **ExoS-Vis chimaera** | | | | | |
| SxVis | —[a] | 378 ± 81 | n.a. | 0.10 ± 0.01 | 4.4 × 10$^{-5}$ |
| SxVis | K-Ras | 114 ± 19 [e] | n.d. | 0.42 ± 0.03 | 6.1 × 10$^{-5}$ |

Calculated from ADP-ribosyltransferase activities using 14-3-3β as a cofactor (except ExoS-Vis chimaera), εNAD$^+$ as a co-substrate, and the indicated substrates (at a concentration near their $K_M$) as an acceptor for the modification Rate data (n = 2) were converted to concentrations of εNAD$^+$ by calibration with εAMP, and fitted to the Michaelis equation. Means ± standard errors are reported
[a]Glycohydrolase activity/automodification
[b]Not applicable
[c]Co-expressed with 14-3-3β
[d]Not determined
[e]In the presence of 30 µM K-Ras

pNIC-H102 (ref. [45]) encoding ExoT target protein CRK$^{1-115}$ (the SH2 domain) with an N-terminal decahistidine tag was contributed by Tomas Nyman (Karolinska Institutet, Protein Science Facility). Oligonucleotide primer sequences are listed in Supplementary Table 4.

**Protein expression and purification**. Generally, proteins were expressed in E. coli strains BL21(DE3) (Stratagene) or T7Express (New England Biolabs) in rich medium (Terrific Broth + 8 g L$^{-1}$ glycerol), by induction of expression during logarithmic growth phase using 0.5 mM isopropyl β-D-1-thiogalactopyranoside (IPTG), followed by bacterial growth overnight at 18 °C. The ExoS$^{233-419}$ constructs (ART domains with C-termini deleted) were co-expressed with 14-3-3β in E. coli strain T7Express lysY (New England Biolabs) by induction with 1 mM IPTG for 3 h at 37 °C. The ExoS:14-3-3β and ExoT:14-3-3β complexes were purified by immobilized metal affinity chromatography (IMAC), employing a hexahistidine tag on the exotoxins, followed by SEC, followed by IEX on HiTrap Heparin columns (GE Healthcare). Purification of ExoS, ExoT, and CRK was by IMAC followed by SEC. The GTPases were purified by IMAC followed by IEX (HiTrap Q-sepharose for K-Ras; HiTrap SP-sepharose for Rac3 and Rnd1). Precipitation of GTPases was minimized by addition of 0.2 mM GTP and 0.2 mM MgCl$_2$ to column fractions. 14-3-3 isoforms were expressed as either hexahistidine fusions and purified by IMAC, or glutathione S-transferase fusions and purified using glutathione sepharose 4B (GE Healthcare). 14-3-3 proteins were further purified by hydrophobic interaction chromatography on Phenyl Sepharose-6 Fast Flow High Sub (GE Healthcare) using a linear gradient of 2.5 to 0 M NaCl in 50 mM sodium phosphate pH 7.0, 1 mM TCEP, followed by SEC. All proteins were concentrated

by ultrafiltration on Vivaspin devices (Sartorius) and stored as aliquots at −80 °C. Protein concentrations were determined using a Nanodrop device (ThermoFisher Scientific) and calculated based on theoretical extinction coefficients. GTPase concentrations (batches containing GTP) were determined using the Bradford assay (ThermoFisher Scientific, 23236) and the albumin standard provided by the manufacturer.

**Protein complex analyses**. Protein complexes were subjected to size exclusion chromatography-right-angle light scattering (SEC-RALS) using a 10/300 Superdex-200 HR column mounted on an ÄKTA Pure chromatography system (GE Healthcare) followed by light scattering analysis using a 8-µl flow cell in a Zetasizer µV instrument (Malvern Panalytical). Light scattering analyses were calibrated using bovine serum albumin dissolved in the experimental buffer and analyzed using OmniSEC software (Malvern Panalytical).

**Protein crystallization and X-ray crystallographic analysis**. Initial work toward a crystal structure is detailed in the Supplementary Methods section. Initial crystals of an ExoS:14-3-3β complex of an estimated molecular weight of 83 kDa were obtained by vapor diffusion in sitting drops at 4 °C. Drops contained 0.15 µl protein (27.6 mg ml$^{-1}$) and 0.15 µl precipitant solution (23% PEG3350, 0.1 M HEPES pH 7.2). Diffraction data were collected from needle crystals at BESSY beamline BL14.1 (wavelength 0.91841 Å) and processed using XDS.[46] Data extended to 3.22 Å resolution, and the space group was C2 (cell parameters a = 134.46 Å, b = 57.31 Å, c = 128.67 Å, β = 112.02°). The asymmetric unit contained two 14-3-3β monomers and one ExoS molecule. The structure was solved by

molecular replacement using PDB entries 2C23 (14-3-3β) and 4XZJ (Vis toxin) as a search model.[21,29] The resulting structure (Supplementary Table 1) served as a molecular replacement template for all following data sets.

Diffraction data were indexed, integrated using XDS,[47] scaled, and truncated using SCALA[48] or XSCALE and the CCP4 suite of programs. BESSY synchrotron diffraction data were processed using XDSAPP.[46] The structures were solved by molecular replacement with PHASER.[49] All structures were refined using phenix. refine[50] or Buster,[51] and model building was done with Coot.[52]

Analysis of the structure showed a crystal packing that allowed the 14-3-3β C-terminus to insert in the active site of ExoS in a neighbor unit cell (Supplementary Fig. 3g, h). We re-designed the co-expression vector, substituting the full length 14-3-3β cDNA with a cDNA coding for a construct truncated after N234. The resulting protein complex yielded crystals in space group C2 that diffracted to 2.34 Å (apo form) and that could be used to determine structures of ligand complexes after soaking (Table 1).

**Enzymatic analyses.** General ExoS assays contained 100 nM ExoS[233–453], 500 nM 14-3-3β, and 25 μM εNAD$^+$ in 20 mM HEPES pH 7.5, 50 mM NaCl, 4 mM MgCl$_2$, and 0.5 mM TCEP. General ExoT assays contained 200 nM ExoT[235–457], 800 nM 14-3-3β, and 100 μM εNAD$^+$. Target proteins were added either at variable concentrations (to determine $K_M^{target}$) or at a concentration near $K_M^{target}$. For negative control reactions, 14-3-3 protein was omitted. Assays were carried out in black flat bottom half area 96-well plates (Greiner 675076) in final volumes of 50 μl. Enzymatic reactions, at ambient temperature (22 °C), were started by addition of εNAD$^+$, and fluorescence was followed over time in a CLARIOstar multimode plater-eader (BMG Labtech) using $λ_{ex}$ = 302/10 nm (filter) and $λ_{em}$ = 410/10 nm (monochromator). All kinetic parameters were determined based on fluorescence increase in the linear time range (typically 10–15 min). Fluorescence was related to concentrations of fluorescent product using serial dilution of 1,$N^6$-etheno-AMP (Jena Bioscience). Fluorescence data were analyzed and kinetic and binding parameters calculated using Prism (Graph Pad Software).

Visualization of substrate protein modification was carried out in essence as above, but with fluo-NAD$^+$ as a co-substrate. Enzymatic reactions were separated by SDS-PAGE on 14% Tris-glycine gels (ThermoFisher XP0014) and imaged on a UV-transluminator before Coomassie staining.

Enzyme inhibition assays (PPI inhibitors) contained either 50 nM ExoS, 50 nM 14-3-3β, 25 μM Rac3, and 40 μM εNAD$^+$, or 80 nM ExoT, 80 nM 14-3-3β, 15 μM CRK SH2 domain, and 40 μM εNAD$^+$. They also contained 3% DMSO and compounds at between 31.25 and 1000 μM. Inhibition was evaluated from ADP-ribosyltransferase rates at different compound concentrations, and EC$_{50}$ values were determined by four parameter curve fitting of the rate data ($N = 2$; $n = 4$).

**Fluorescence anisotropy assay of ExoS–14-3-3 interaction.** GFP-ExoS protein constructs were pre-incubated with 14-3-3 proteins in 20 mM HEPES, 150 mM NaCl, 1 mM TCEP, pH 7.5, and parallel and perpendicular GFP-fluorescence intensities were determined in a CLARIOstar multimode plate reader (BMG Labtech) using $λ_{ex}$ = 482/16 nm, $λ_{em}$ = 530/40 nm, and a bandpass of 504/10 nm. Anisotropy was calculated according to the following equation:

$$r = \frac{I_{||} - I_\perp}{I_{||} + 2I_\perp}, \tag{1}$$

where $r$ is anisotropy, $I_{||}$ is the fluorescence intensity parallel to the excitation light, and $I_\perp$ is the fluorescence intensity perpendicular to the excitation light. Apparent binding constants were calculated by fitting anisotropy to a quadratic-binding equation using Microcal Origin:

$$r = r_0 + (r_\infty - r_0)\left(\frac{P_T + L_T + K_d - \sqrt{(P_T + L_T + K_d)^2 - 4P_T L_T}}{2}\right), \tag{2}$$

where $r$ is the anisotropy of the population of GFP-ExoS in solution, $r_0$ is the initial anisotropy (in the absence of 14-3-3), $r_\infty$ is the anisotropy of GFP-ExoS at saturating [14-3-3], $P_T$ is the total [14-3-3], $L_T$ is the total GFP-ExoS, and $K_{d,app}$ is the apparent binding affinity of GFP-ExoS for 14-3-3.

**Thermal aggregation assay by DLS.** ExoS[233–453] (10 μM) and 14-3-3β (variable) were pre-incubated in 20 mM HEPES, 300 mM NaCl, 10% glycerol, 1 mM TCEP, pH 7.5. Samples were cleared of aggregates in a table top centrifuge (17,500 × $g$; 30 min, 4 °C). DLS analyses were performed on 50 μl of the supernatant in a quartz cuvette using a detector at 90° from the incident laserbeam in a Zetasizer μV instrument (Malvern Panalytical). Protein stability was assessed using a temperature gradient from 25 to 70 °C. Z-average size was plotted against temperature and the aggregation onset temperature was determined by extrapolating the linear section of the transition, at its highest slope, backward to its intersection with the baseline at low temperatures.

**Assay of protein toxicity in yeast.** ExoS[233-419], ExoS[233-435], ExoS[233-453] sequences were amplified by PCR and integrated downstream the galactose-

inducible GAL1-10 promoter in the yeast integrative plasmid vector YIplac211. Correct integration was controlled by DNA sequencing. Plasmids were linearized using the StuI restriction enzyme (Roche), and integrated at the URA3 locus in the budding yeast Saccharomyces cerevisiae W303 using lithium acetate-mediated transformation.[53] The genotypes of the resulting strains are listed in Supplementary Table 5. To evaluate the viability of yeast strains expressing the different ExoS variants, a volume equivalent to OD$_{600}$ = 3 of mid-log phase cells grown in YEP media supplemented with 2% glucose were harvested using centrifugation. Resulting cell pellets were resuspended in 1 ml sterile water, and 3 μl of 5-fold serial dilutions were spotted on YEP plates supplemented with 2% glucose (expression off), or with 2% raffinose and 2% galactose (expression on). Images shown in Fig. 3e were taken after 24 h incubation at 30 °C.

**Assay of protein toxicity in E. coli.** Cells of strain T7Express lysY (New England Biolabs) were transformed with expression plasmids and grown on agar overnight. Liquid cultures (Lucia Broth) were prepared and allowed to grow at 37 °C to saturation. New liquid cultures (Terrific Broth) were inoculated and allowed to grow at 37 °C to OD$_{600}$ of 1.0–1.2. Cultures were diluted to OD$_{600}$ of 0.1 using either Terrific Broth or Terrific Broth supplemented with 1 mM IPTG. Serial dilutions were prepared, of which 10 μl were spotted on either agar or agar supplemented with 1 mM IPTG. Plates were imaged after overnight incubation at 37 °C.

**Quantification and statistical analysis.** The statistical details, number of experiments and statistical analyses, are described in the figure legends or in the Methods in all relevant cases. Software used for quantification and data analysis is indicated under experimental details in Methods.

## Data availability

All crystal structures described in this work have been deposited to the Protein Data Bank (PDB) under the accession numbers codes 6GN0, 6GN8, 6GNJ, 6GNK, and 6GNN. All other data supporting the findings of this study are available upon reasonable request from the corresponding author.

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

## Acknowledgements

We thank Pontus Aspenström, Åke Forsberg, Tomas Nyman, Charlotta Sundin, and Yngve Östberg for contributing cDNA, and the Protein Science Facility at Karolinska Institutet/SciLifeLab for molecular cloning. We thank the beamline staff at the synchrotron radiation facilities BESSY (Berlin, Germany), ESRF (Grenoble, France), and Diamond (Didcot, UK) for excellent support. This work was supported by the Swedish Foundation for Strategic Research (SB12-0022; to M. Elofsson and H.S.), the Swedish Research Council (2012-2802 to M. Elofson; 2015-4200 to C.B., and 2015-4603 to H.S.), the IngaBritt and Arne Lundbergs Research Foundation (403 and 2015-089 to H.S.), National Institutes of Health (R01 Grant GM097348 to E.M.D.L.C.). E.V.W. was supported by a National Science Foundation Graduate Research Fellowship (DGE-1122492) and K.N. by a Wenner-Gren Foundation fellowship.

## Author contributions

Conceptualization: T.K., P.H., and H.S. Methodology: T.K., P.H., A.F.P., E.V.W., E.M.D. L.C., C.B., and H.S. Investigation and validation: T.K., P.H., A.F.P., S.M., M. Ebrahimi, N. P., A.N., E.L., A.G.T., and H.S. Resources: M.L., R.C., and M. Elofsson. Formal analysis: T.K., E.V.W., K.N., and E.M.D.L.C. Writing—original draft preparation and project administration: H.S. Writing—review & editing: all authors. Visualization: T.K. and H.S. Supervision and funding acquisition: M. Elofsson, E.M.D.L.C., C.B., and H.S.
