## [Peer Review File · Nature Communications]

Reviewers' comments:

Reviewer #1 (Remarks to the Author):

This is an interesting and well prepared manuscript which analyzes the effect of 14-3-3 proteins on Pseudomonas Exotoxins S and T by crystallography. Based on the solved structures of co-complexes the authors performed further biochemical analysis and show that the main role of the 14-3-3-ExoS/T interaction is to prevent aggregation. This may be especially relevant during delivery by passage through the type 3 secretion apparatus and refolding of the ADP-ribosyltransferase. The data generated are convincing, complete and important for the understanding of the action of bacterial effectors. I fully recommend the manuscript for publication in nature communications.

Reviewer #2 (Remarks to the Author):

The manuscript by Karlberg et al. reports structural-functional analysis of the interesting protein-protein interaction between the phosphopeptide-binding 14-3-3 proteins and either of two Pseudomonas exotoxins, ExoS and ExoT, not requiring phosphorylation of the latter, which is otherwise common for 14-3-3/partner interactions. The authors claim that their analysis of 14-3-3/ExoS(T) interaction, including several novel crystal structures of the complexes at different stoichiometries and complexes with small molecules, provides understanding of the mechanism of the action of these toxins, that is not associated with the effect of 14-3-3 directly on the enzymatic activity of the exotoxins, but with its chaperone-like functioning on ExoS/T using hydrophobic interactions.

This topic certainly represents interest to the community focused on molecular basis of host-pathogen interactions and also to the (wide) 14-3-3 field, especially because mainly 14-3-3/phosphopeptide complexes and only a few complexes with longer fragments/domains of 14-3-3 partners have been possible to co-crystallize and solve so far. However, despite leaving an overall positive impression, the paper certainly needs more work to convince the reader that the action of 14-3-3 proteins in the ExoS/T case is indeed determined by the chaperone-like activity of the former. In particular, self-association and aggregation propensity of ExoS and ExoT in the absence and presence of 14-3-3 needs to be carefully analyzed. Please refer to the recent literature rather strictly discriminating between the "classical" target binding and the chaperone-like activity of 14-3-3s. Also, can it be that, by forming tight complexes, ExoS/T simply sequester the important and universal regulator of the host cell, 14-3-3, precluding it from performing normal functions? What are typical concentrations of these toxins in the cell? Please discuss this possibility as well.

I have further questions and recommendations, both major and less critical, that may be used to improve the paper before resubmission.

Main text.

1. Introduction seems to lack the description of the main mechanism of toxicity mediated by ADP-ribosylation, K-Ras inactivation in particular. Is ADP-ribosylation itself dangerous to the host cells? The host cells have their own ADP-ribosylation systems, how do these interfere with those derived from pathogens like Pseudomonas? Please set a good "biological" scene in the introduction as the paper really pretends for a wide audience.

2. Lines 36-38. "Two of these, exotoxins S and T (ExoS; ExoT) are homologous enzymes consisting of an N-terminal GTPase activating protein (GAP) domain and a C-terminal ADP-ribosyltransferase (ART) domain." Please provide the level of homology (% identity between S and T).

3. Lines 41-43. "The activities of the ART domains are directed toward a more diverse set of proteins. ExoS targets Ras- and Rho-family GTPases and ezrin/radixin/moesin (ERM) proteins, and ExoT targets the adaptor molecules, CT10 Regulator of Kinase (CRK) and CRK-like (CRKL)." No reference is provided. Please expand the description by explaining also the consequences of targets modification by ExoS/T and the essence of ADP-ribosylation - why is it needed and how can it affect targets of an ADP-ribosylase?

4. "Grove" (vs. groove) spelling seems incorrect.

5. There are significant repetitions of the methods in the main text and SI (crystallization and SEC-RALS)

6. Numbers characterizing resolution (e.g., 3.2 and 3.24Å) are used inconsistently.

7. Figure 1. The SDS-Page of the load and eluted trimer and tetramer should be presented to show 1) the purity and 2) stoichiometry of the complexes.
8. Fig. 1d. An arrow with two arrowheads connecting F354 in both conformations might be helpful to highlight putative conformational changes associated with the ligand binding.
9. Fig. 1. Indicate that ExoS and T constructs on panels c, e, h, g are shown by a gradient from the N (blue) to C (red) terminus.
10. Connection of the fragments bound in the grooves by a linker (residues 432-443) is recommended (by a curved dashed line?) for the trimeric complex (panel c).
11. RMSD of the tetrameric structures with ExoS or T is recommended to be included in the text (maybe separately, using superposition with or without considering 14-3-3 dimers) to clarify the similarity of the structures with the two homologous toxins.
12. Line 126. The reference to Fig. 2c is wrong, on this panel only binding curves are presented.
13. Fig. 2b is not very explanatory and is difficult to understand.
 - 13.1. Which part of the structure, presented on panel a of the same figure, panel 2b does correspond to? (electrostatic potentials are not very helpful there).
 - 13.2. What is this structural element? B-strand? Loop? Maybe semitransparent surface representation (of ExoS) with cartoon representation of structural elements and highlighting of the region of interest by a dashed circle on panel a would be more helpful. The lower part of the panel a is not clear, especially because the view is reversed compared to that on panel b (E332 is on top in a and in the bottom on panel a, low part). Unfortunately, "two key interactions indicated" are not sufficient to understand the picture.
 - 13.3. Arrows on panel a should indicate rotation, however the idea is not clear (what is rotated? by what degree? towards what direction?).
14. Fig. 2c. In the legend it is said that only dose-dependent inhibition of ExoS is presented, while the curves for both ExoS and ExoT are shown.
15. Fig. 3d. Fluorescence polarization data have significant problems that need to be addressed.
 - 15.1. Why are different scales presented along the X axis?
 - 15.2. The difference between "233-453" and "233-435" cases are barely seen, considering also scattering of the data.
 - 15.3. Why does the extended scale on the second plot show gradual decrease of anisotropy after a region of saturation (not shown on the other two plots probably because of the different scale)?
 - 15.4. I doubt that the correct estimation of the K_d is possible using the data on the third plot, without reaching any saturation. Without reaching saturation on Fig 3d (3d plot) it is possible to say that the app. K_d is not less than 100 nM, but may be 150 or 1500 nM. The same holds for the supplementary fig. s6.
 - 15.5. Although the effect of the LDLA box truncation are somewhat intuitive and expected, the data presented are not really convincing. According to the current knowledge, binding in the amphipathic groove of 14-3-3 indeed confers relatively high affinity (micromolar range) and is considered the primary interaction, whereas binding in the absence of such primary binding is considered much less affine, at least no clear example exists. The authors claim sub-micromolar affinity only at their "novel" interface, i.e., outside the grooves of 14-3-3, which is potentially interesting but also a somewhat strong claim. An alternative assay is highly recommended, to quantify the affinity of this binary interaction and compare with any known 14-3-3 interaction, which would be a useful addition to the statement that the novel binding interface is really tight. Please, use ITC or at least native PAGE to quantify the 14-3-3 interaction with 233-419 ExoS (the weakest claimed in Fig. 3d, but much stronger than for other known 14-3-3 binders!).
 - 15.6. Please either provide the exact equation used to calculate K_d s (with the description of parameters) or a reference (less preferred) in the corresponding part of the MMs.
16. Lines 175-177. "This positioning interaction pattern were both virtually identical in previous crystal structures of ExoS derived peptides in complex with 14-3-3 proteins (Fig. 3).10,12" The statement is unclear, please rephrase.
17. I wonder why the interfaces with ExoS and -T at the "novel" binding site (with a8 and a9 helices of 14-3-3) are not compared in the paper – the interaction is only analyzed in detail using ExoS as an example.
 - 17.1. How conserved are the local contacts in these two cases (overall folds of the ExoS and ExoT complexes with 14-3-3b look similar from Fig. 1)?
 - 17.2. Is it possible to confirm the universal principle of binding in this outside binding site for all 1433 dependent toxins, based on the differences and similarities of their sequences in the region involved in the novel binding? This comparative analysis seems important for the logics of the paper and for the putting forward the mechanism of toxin activation and 14-3-3 dependence.
 - 17.3. Along these lines, 14-3-3 independent toxin Vis (lacking LDLA boxes 1 and 2) still has considerable homology with the ExoS and ExoT toxins (suppl. Fig. 5) and therefore is expected to also interact at the "novel" binding site on C-terminal helices of 14-3-3. Is this the case?

Otherwise, it would be informative to analyze local differences of Vis (pdb 4xzj) and ExoS/T to rationalize the inability of Vis to bind 14-3-3.

18. As said before, the hypothesis about the chaperoning function of 14-3-3s on ExoS, which is claimed to be aggregation prone, is rather weak as the experimental verification of this important statement is superficial. The main evidence is DLS data showing the modest stabilizing effect (by only ~4C) of 14-3-3 on "Tagg" of ExoS. This part has technical problems and is incomplete.

18.1. Technically, how was Tagg calculated? This is a rarely used parameter to characterize aggregation (in contrast to denaturation) because it is often difficult to identify the region of aggregation curve saturation, which would be necessary to derive the half-transition parameter, because the last part of the aggregation curves are typically recorded either in already turbid solutions with extremely high fluctuations of the signal or in solutions with no turbidity, but thus in the absence of saturation whatsoever. This seems exactly the case for the curves recorded in the presence of various 14-3-3 concentrations, for example, the highest ratio (2.0) curve shows no saturation and therefore no T0.5 parameter can be correctly determined by sigmoidal curve fitting, as it is stated in MM section. Much better and robust parameter would be temperature at which aggregation starts to occur (possible to determine in the absence of saturation). This could be readily determined by extrapolating the rising part of the thermal curve backwards to the X axis (will be something like 45C in the case of ExoS alone from Fig. 4i).

18.2. Next problem is that ExoS starts to aggregate at only 45C, which means it is a rather stable protein, at least under physiological conditions, not necessarily aggregation-prone.

18.3. Another issue concerns the fact that the DLS data for only ExoS with two LDLA boxes are presented and therefore no conclusion can be drawn for the ExoS ART domain (lacking LDLA boxes). To address this, the same analysis should be done for the individual ART domain (in the absence or presence of various 14-3-3 concentrations), which would tell whether 14-3-3 affects the aggregation behaviour of ExoS in the absence of the binding at the amphipathic groove(s).

18.4. Next important thing that is currently lacking is the analysis of the oligomeric status of native (non-aggregated) ExoS (both, containing and lacking LDLA boxes) depending on protein concentration, which would support the categorical conclusion about the aggregation propensity of ExoSs. The doubt is whether in in vivo situation these toxins indeed suffer from this speculative aggregation propensity; overexpression of the recombinant proteins often leads to their increased aggregation, not necessarily because of their intrinsic propensity to aggregate.

18.5. Ideally, if the aggregation propensity of ExoS/T is confirmed under more natural conditions, the nature of this propensity is good to address to distinguish between self-association of the native-like protein particles (due to their enhanced surface hydrophobicity) or aggregation accompanying their denaturation.

18.6. Is it possible to speculate that 14-3-3-independent ART-containing toxins (lacking LDLA boxes) are in fact dependent on 14-3-3 chaperone-like activity?

18.7. It is strange that the hypothesis about the chaperoning activity of 14-3-3s comes "out of the blue", without being even mentioned in introduction or earlier sections of the study. In fact, chaperone-like activity is a relatively poorly characterized, moonlighting activity of 14-3-3 proteins, different in many aspects from the main phosphopeptide-binding activity (ref 5 of the paper). Despite the current paper uses the terms and hypothesizes that this activity of 14-3-3 with respect to ExoS/T takes place, it does not properly introduce the chaperone function of 14-3-3s in the paper. Please, discuss the problem and the role for hydrophobic clusters in aggregation and chaperone activity more, referring to the research you already cite (ref. 5).

19. Discussion (ca 250 words). I don't know the purpose of these parentheses, but the actual discussion is >700 words.

20. Discussion of the "novel binding site".

20.1. Line 120. It is stated that the novel site can be sub-divided into two parts. It is not immediately clear from the Figure 2 what authors mean, please consider marking those two sub-parts in the figure.

20.2. I do not agree that the binding site is really novel and that it only slightly overlaps with the sites found, for example, in 14-3-3 complexes with HSPB6 (pdb 5ltw). Supplementary fig. s12c already contradicts this statement, showing a nice overlap of the binding surfaces, comparably with that in the case of trehalase (pdb 5n6n) – fig. s12a.

20.3. I especially disagree with the novelty of the idea that "In a development of pharmacological interventions involving the novel exotoxin binding site" and the originality of the statement that "it will be important to address the question of whether other proteins interact at this site". In fact, it has been recently discussed that i) interaction of partners outside the amphipathic grooves is quite often, ii) that $\alpha 8$ - $\alpha 9$ helices often take part in such associations and iii) that, considering the lower affinity and higher variability of binding using $\alpha 8$ - $\alpha 9$ helices of 14-3-3 (compared to the binding at the amphipathic grooves), drug targeting of such secondary interactions is promising (PMID: 29180038). Please, cite this preceding work which falls exactly in the topic and discussion of your

paper. In light of these facts, targeting of the outer surface of $\alpha 8$ - $\alpha 9$ helices of 14-3-3 to prevent toxin binding may also be challenging because other (vital) interactions besides those with toxins may be inhibited, therefore the statement "This is especially important as compounds binding to an exotoxin specific site would circumvent the off-target effects expected of inhibitors that bind in the amphipathic groove of 14-3-3 proteins. 29". (lines 307-309) is rather applicable to both interactions, at the amphipathic groove and outer surface of 14-3-3.

21. Crystallographic part has some inaccuracies and minor problems that need to be addressed.

21.1. It is a complete mystery what was the purpose of obtaining the co-crystal structures of ART domains and the STO1101 compound as it is only stated (lines 100-104) that the structures were obtained and this is not discussed in the paper at all.

21.2. Considering the crystal structure with STO1101, it has too low resolution (3.2 or 3.8 Å only) to confidently place the inhibitor and reveal the contacts (as e.g. shown in Fig S2c and d). Please show the omit electron density map for the bound inhibitor to support the claims.

21.3. It is not clear why the heterotrimeric complex (83 kDa) was first solved using datasets from Bessy (but not refined – first column of the Table I is half empty) and then – using data collected in Diamond, and both datasets are presented in Table I. The different unit cell parameters (primarily beta-angle – 112 and 126 degrees) could be noticed, doubting whether the two crystal forms were completely isomorphic and the same. Please, either remove the first column (and its description in MM) or refine and show the full column in Table I.

21.4. The last column of Table I indicates that this structure is considered refined and even about to be (or already) deposited with the PDB, despite the text states that the structure had poor resolution and the refinement was abandoned. Of course at this low resolution the refinement is challenging, but the R_{free} of 40 cannot be considered good. Please, either try refining the structure more or remove the second part of the last column in Table I. Otherwise the structure at 3.8Å and R_{free} of 40% will not be informative and useful.

21.5. The scheme on Fig. 1a is misleading because from it it follows that truncation of several residues from the C-terminus of 14-3-3 β results in switching of the stoichiometry of its complexes with ExoS (from heterotetramer to heterotrimer).

21.6. It remained unclear from the paper, what was the determining factor of the formation of either trimeric and tetrameric complexes with 14-3-3. Was it just a spontaneous event and these complexes formed spontaneously and then could be resolved by heparin chromatography? Please, make this point more clear.

Supplement.

Line 61. Reference to Fig. S2 should be to S3. Otherwise, change S2 and S3.

Line 67. Reference to Fig. S1, should be to S3.

Fig. S1b. δ isoform is indicated, supposed to be γ instead.

Fig. S3. Panels a-d. OY axes labeled inconsistently (Abs280 and UV), why so?

What could explain different toxicity of ART domains of ExoS and ExoT for E.coli cells, which resulted in the ability of authors to obtain only ExoTwt, whereas ExoS needed mutations in the active site to overcome toxicity?

Fig. S3e. Could some different tint of purple/pink/violet used to color 14-3-3 dimer in the tetrameric symmetry mate? Now it is difficult to understand where there are 14-3-3 dimer and ART domain monomers.

Legend to Fig. S5. Lines 125-127. "14-3-3 dependent".. the word "toxins" seems to be missing
Line 137. "or LDLA-boxes 1 and 1" - should be 1 and 2?

Fig. S8. Please comment on the presence of multiple bands on the ExoS lane? Does it reflect poor purity of the sample?

Fig. S9. Why there is no protein present in the first band (ExoS)? Why is the 14-3-3 band to the right of the standards fluorescent (only in fig s9 out of s8-s11)? Does it also serve as a substrate for ADP-rib reaction along with K-Ras (only under these specific conditions)? Please comment on this.

Fig. S12. Legend. A bunch of mistakes "14-4-4". Similar attempt to overlay existing complexes and to identify binding regions outside the amphipathic grooves has already been done recently in (PMID: 29180038), please cite this work.

Fig. S13. Please indicate the position of the A-loop, which is discussed in the legend on structural pictures.

Nikolai Sluchanko

Reviewer #3 (Remarks to the Author):

Summary:

This study advances understanding of how ExoS and ExoT ADP ribosyltransferase activity is modulated by 14-3-3 proteins. The authors provided cocrystals of adequate resolution to support their findings, and then tested key interactions of residues by mutagenesis or truncations. They identified a new region of interaction between ExoS/T and 14-3-3b in the second LDLA box at the C-terminus in one co-crystal structure, show small reduction in affinity to 14-3-3b, and speculate on implications from this. They are able to construct a 14-3-3b-independent ExoS by making a fusion with Vis toxin. Given the retained affinity for 14-3-3b to ExoS lacking both LDLA boxes, they investigate whether enzymatic activity is still possible when both proteins are co-expressed. The overall findings advance the field in two ways: 1. they have provided a complete ExoS ART domain structure while also confirming partial structures of fragments from previous publications, and 2. They show that the mechanism of activation by 14-3-3b appears to be through alleviating aggregation. Both are sufficiently novel contributions, and of importance to the bacterial protein toxin field and *P. aeruginosa* pathogenesis. The proposed mechanism of 14-3-3 as a chaperone for ExoS rather than inducing a conformational change to the active site may be of particular interest to a wider audience.

Specific comments:

1. The activity of the ExoS-Vis fusion protein is interesting and unexpected, even with its reduced activity. Is the Vis C-terminus fusion sufficient to alleviate aggregation, too?
2. One of the key findings shows that ExoS 233-419 (lacking LDLA motifs) is still active when co-expressed. It is not immediately clear how the co-expressed proteins could be obtained for this experiment in the text given that they are toxic to *E. coli*; the methods indicate a shorter induction with increased IPTG, which presumably got around this problem. If that is correct, a short statement preceding line 256 in the text could avoid confusion.
3. Related to point 2, this particular result (catalytically active ExoS 233-419 when co-expressed with 14-3-3b) seems unexpected given ExoS 233-419 is not toxic in yeast in Figure 3e. Perhaps the authors can hypothesize on why ExoS 233-419 is not toxic in yeast, or has no partial effect, even though it might be catalytically active.
4. A critical component of the authors' model depicting 14-3-3b as a chaperone depends on a single dynamic light scattering experiment and is only otherwise supported by their observation of inclusion bodies. Perhaps this finding could be strengthened if a negative control, such as an irrelevant protein, is also shown in addition to the titration of 14-3-3b.
5. The authors are a bit sparse on citations of previous work in their introduction. For example, they do not cite discovery of the *P. aeruginosa* T3SS nor any of the work identifying targets of ExoS and ExoT. However the writing is of good quality and will be easy to follow by those in the field.

To summarize our activities during the revision of our manuscript; we have generated experimental data on the thermal aggregation of ExoS measured by DLS (some of them requested by reviewers 2 and 3; others to improve statistics or in support of conclusions), and have made a number of suggested changes to figures and text.

All crystal structures that constitute part of the results have been deposited with the Protein Data Bank (entry codes are indicated in the manuscript) and will be ready for release within days.

Please see our detailed response in blue below.

Reviewers' comments:

Reviewer #1 (Remarks to the Author):

This is an interesting and well prepared manuscript which analyzes the effect of 14-3-3 proteins on Pseudomonas Exotoxins S and T by crystallography. Based on the solved structures of co-complexes the authors performed further biochemical analysis and show that the main role of the 14-3-3-ExoS/T interaction is to prevent aggregation. This may be especially relevant during delivery by passage through the type 3 secretion apparatus and refolding of the ADP-ribosyltransferase. The data generated are convincing, complete and important for the understanding of the action of bacterial effectors.

I fully recommend the manuscript for publication in nature communications.

We thank the reviewer for the positive comments.

Reviewer #2 (Remarks to the Author):

The manuscript by Karlberg et al. reports structural-functional analysis of the interesting protein-protein interaction between the phosphopeptide-binding 14-3-3 proteins and either of two Pseudomonas exotoxins, ExoS and ExoT, not requiring phosphorylation of the latter, which is otherwise common for 14-3-3/partner interactions. The authors claim that their analysis of 14-3-3/ExoS(T) interaction, including several novel crystal structures of the complexes at different stoichiometries and complexes with small molecules, provides understanding of the mechanism of the action of these toxins, that is not associated with the effect of 14-3-3 directly on the enzymatic activity of the exotoxins, but with its chaperone-like functioning on ExoS/T using hydrophobic interactions.

This topic certainly represents interest to the community focused on molecular basis of host-pathogen interactions and also to the (wide) 14-3-3 field, especially because mainly 14-3-3/phosphopeptide complexes and only a few complexes with longer fragments/domains of 14-3-3 partners have been possible to co-crystallize and solve so far. However, despite leaving an overall

positive impression, the paper certainly needs more work to convince the reader that the action of 14-3-3 proteins in the ExoS/T case is indeed determined by the chaperone-like activity of the former. In particular, self-association and aggregation propensity of ExoS and ExoT in the absence and presence of 14-3-3 needs to be carefully analyzed. Please refer to the recent literature rather strictly discriminating between the “classical” target binding and the chaperone-like activity of 14-3-3s. Also, can it be that, by forming tight complexes, ExoS/T simply sequester the important and universal regulator of the host cell, 14-3-3, precluding it from performing normal functions? What are typical concentrations of these toxins in the cell? Please discuss this possibility as well.

We thank the reviewer for the very detailed criticism, which we have found very useful.

We are aware that the chaperone role of 14-3-3 proteins in the activation of ExoS and similar toxins is not fully established with our results. Rather, our results provide evidence for a new working model of the activation of ExoS and other 14-3-3 activated toxins. We believe that this distinction is even clearer in the new version of the manuscript.

The distinction between the “classical” target binding mode and a binding mode mainly mediated by hydrophobic interactions is made here by using two primary types of exotoxin constructs: Those that contain the C-terminus with the LDLA motifs (which bind 14-3-3 mimicking the “classical” mode without being phosphorylated) and those that lack these motifs (being restricted to binding via the novel hydrophobic site that we describe here). In the new version of the manuscript, we present new data that show the effect of 14-3-3 on aggregation of the latter construct. The results show that 14-3-3 stabilizes also the ExoS construct that lacks the C-terminal phosphopeptide site binding tail. Furthermore, we show that the PPI inhibitor STO1704 counteracts the 14-3-3 mediated stabilization of ExoS in the aggregation assay. Together, these data support our interpretation that the chaperone-like activity of 14-3-3 on ExoS is mediated by the novel binding site. The new data are presented in the Supplementary Information, Supplementary Fig. S14 and Table S3 of the new version.

With the “14-3-3 sequestering model”, the reviewer raises an interesting and important question. As we allude to in the manuscript, we believe that this may indeed be an aspect of toxin regulation; but have insufficient experimental evidence to understand it fully. The number of ExoS molecules required for *Pseudomonas* to gain access to a host cell, or the number of copies to elicit toxic effects in a host cell, have not been established. (It has been estimated, by quantitation of GFP fluorescence after replacement of endogenous ExoS with a GFP-ExoS fusion gene, that ~100 molecules of ExoS cause apoptosis, cell rupture and bacterial dissemination: PMID 26090668.) Nevertheless, it is reasonable to assume that single, or very few, copies of ExoS should elicit an effect that is favorable for the bacteria, i.e., allow access to a replication competent niche. At those copy numbers one would expect that the effect caused directly by sequestration of the multi copy number 14-3-3 proteins is negligible. Instead, we believe that 14-3-3 sequestration by the toxins might constitute part of a feedback loop that may prevent too many toxin copies to be active inside a host cell. We expect such a mechanism to exist because the bacteria have no use for the host cell unless it survives over sufficient time to allow bacterial propagation. We consider work to validate such a model outside the scope of the current manuscript, because it would require consideration of (1) full length toxins and (2) their functioning in a cellular context.

I have further questions and recommendations, both major and less critical, that may be used to improve the paper before resubmission.

Main text.

1. Introduction seems to lack the description of the main mechanism of toxicity mediated by ADP-ribosylation, K-Ras inactivation in particular. Is ADP-ribosylation itself dangerous to the host cells? The host cells have their own ADP-ribosylation systems, how do these interfere with those derived from pathogens like *Pseudomonas*? Please set a good “biological” scene in the introduction as the paper really pretends for a wide audience.

We have re-written the first paragraph of the paper to summarize the requested information.

2. Lines 36-38. “Two of these, exotoxins S and T (ExoS; ExoT) are homologous enzymes consisting of an N-terminal GTPase activating protein (GAP) domain and a C-terminal ADP-ribosyltransferase (ART) domain.” Please provide the level of homology (% identity between S and T).

The degree of homology between the two exotoxins has been added.

3. Lines 41-43. “The activities of the ART domains are directed toward a more diverse set of proteins. ExoS targets Ras- and Rho-family GTPases and ezrin/radixin/moesin (ERM) proteins, and ExoT targets the adaptor molecules, CT10 Regulator of Kinase (CRK) and CRK-like (CRKL).” No reference is provided. Please expand the description by explaining also the consequences of targets modification by ExoS/T and the essence of ADP-ribosylation – why is it needed and how can it affect targets of an ADP-ribosylase?

This description has been expanded and the original references have been introduced.

4. “Grove” (vs. groove) spelling seems incorrect.

This mistake has been corrected throughout.

5. There are significant repetitions of the methods in the main text and SI (crystallization and SEC-RALS)

We have formatted the methods section according to journal style, but kept some background information (targeting specialists) in the Supplementary Methods. Some repetition has been kept in favor of readability.

6. Numbers characterizing resolution (e.g., 3.2 and 3.24A) are used inconsistently.

This has been changed throughout.

7. Figure 1. The SDS-Page of the load and eluted trimer and tetramer should be presented to show 1) the purity and 2) stoichiometry of the complexes.

This has been added as Supplementary Fig. 4.

8. Fig. 1d. An arrow with two arrowheads connecting F354 in both conformations might be helpful to highlight putative conformational changes associated with the ligand binding.

This change has been made.

9. Fig. 1. Indicate that ExoS and T constructs on panels c, e, h, g are shown by a gradient from the N (blue) to C (red) terminus.

This information has been added to the figure legend.

10. Connection of the fragments bound in the grooves by a linker (residues 432-443) is recommended (by a curved dashed line?) for the trimeric complex (panel c).

We understand the rationale behind this comment (improved clarity); however we prefer to show only those secondary structural elements for which electron density was observed. We believe that the origin of the C-terminal helix is sufficiently clear by the labels given the sequence range and the panel title (“heterotrimer”).

11. RMSD of the tetrameric structures with ExoS or T is recommended to be included in the text (maybe separately, using superposition with or without considering 14-3-3 dimers) to clarify the similarity of the structures with the two homologous toxins.

The structural similarity has been added (first paragraph of results section). As the ExoS monomer of the relevant structure was ligand free, the secondary structural elements around the active site are not in the closed conformation, making it necessary to explain to more detail why the overall RMSD for the complex is relatively high in this comparison. We have done that by adding Supplementary Fig. 5, which presents stereo views of the superpositions of complexes and ART domains alone. We thank the reviewer for pointing this out.

12. Line 126. The reference to Fig. 2c is wrong, on this panel only binding curves are presented.

This has been corrected.

13. Fig. 2b is not very explanatory and is difficult to understand.

13.1. Which part of the structure, presented on panel a of the same figure, panel 2b does correspond to? (electrostatic potentials are not very helpful there).

13.2. What is this structural element? B-strand? Loop? Maybe semitransparent surface representation (of ExoS) with cartoon representation of structural elements and highlighting of the region of interest by a dashed circle on panel a would be more helpful. The lower part of the panel a is not clear, especially because the view is reversed compared to that on panel b (E332 is on top in a and in the bottom on panel a, low part). Unfortunately, “two key interactions indicated” are not sufficient to understand the picture.

13.3. Arrows on panel a should indicate rotation, however the idea is not clear (what is rotated? by what degree? towards what direction?).

The figure and legend have been extensively modified, taking the reviewer’s requests into account.

14. Fig. 2c. In the legend it is said that only dose-dependent inhibition of ExoS is presented, while the curves for both ExoS and ExoT are shown.

This has been corrected (now 2d).

15. Fig. 3d. Fluorescence polarization data have significant problems that need to be addressed.

15.1. Why are different scales presented along the X axis?

We have re-drawn the graphs; all present the same scale on the X-axis in the new version.

15.2. The difference between “233-453” and “233-435” cases are barely seen, considering also scattering of the data.

The conclusion that the terminal LDLA sequence has little effect on overall 14-3-3 binding is the point we wish to make, and is also in agreement with the remainder of the data. The values are provided by the unbiased best fit of the data to a quadratic binding equation; but we agree with the reviewer that the underlying data do not warrant concluding that there is a significant difference in 14-3-3 binding between these two constructs. To reflect this, we have changed the text in the results section as well as the figure annotation.

15.3. Why does the extended scale on the second plot show gradual decrease of anisotropy after a region of saturation (not shown on the other two plots probably because of the different scale)?

The reviewer astutely identifies a gradual decrease in anisotropy beyond the saturation region. The reason for this decrease is currently unknown and may be a consequence of the extended data range, as suggested. However, the inability to access the same regime in the other three graphs had led us to fit that data with a comparable range for all four constructs, such that the $K_{d,app}$ of all constructs reflects similar [14-3-3], such that any differences are likely due to interactions with ExoS as opposed to 14-3-3-related interactions.

15.4. I doubt that the correct estimation of the K_d is possible using the data on the third plot, without reaching any saturation. Without reaching saturation on Fig 3d (3d plot) it is possible to say that the app. K_d is not less than 100 nM, but may be 150 or 1500 nM. The same holds for the supplementary fig. s6.

We thank the reviewer for their attention to detail and appreciation of the binding curves presented. We were aware of this issue, and used “sub-micromolar affinity” in our description of the results. We have now changed all instances in the text and figure annotations to indicate the ability of the fits to provide lower limits of $K_{d,app}$ at a minimum.

15.5. Although the effect of the LDLA box truncation are somewhat intuitive and expected, the data presented are not really convincing. According to the current knowledge, binding in the amphipathic groove of 14-3-3 indeed confers relatively high affinity (micromolar range) and is considered the primary interaction, whereas binding in the absence of such primary binding is considered much less affine, at least no clear example exists. The authors claim sub-micromolar affinity only at their “novel” interface, i.e., outside the grooves of 14-3-3, which is potentially interesting but also a somewhat strong claim. An alternative assay is highly recommended, to quantify the affinity of this binary interaction and compare with any known 14-3-3 interaction, which would be a useful addition to the statement that the novel binding interface is really tight. Please, use ITC or at least native PAGE to quantify the 14-3-3 interaction with 233-419 ExoS (the weakest claimed in Fig. 3d, but much stronger than for other known 14-3-3 binders!).

We agree that alternate methodologies increase the certainty of the binding parameters, and appreciate the reviewer's concern for the integrity of the data. The GFP anisotropy signal appears to reliably report on [14-3-3]-dependent changes in the fluorescent ExoS species. We do not believe that native gel assays suffice as an alternative to this quantitative method. We have been unable to conduct ITC measurements, owing to limiting ExoS protein, in particular for the 233-419 construct. We have been unable to verify the binding affinities using surface plasmon resonance (Biacore) for any ExoS protein construct, which we attribute to surface binding effects (several chip chemistries and several tags on the proteins were used). Thus, to date, we cannot provide a second estimate of the binding affinities for these proteins. However, we consistently used broad terms (e.g., "sub-micromolar affinity" instead of "Kd of 100 nM") to reflect a residual uncertainty; and we believe that with the changes we introduced in the text (see previous comment), it will be clear to readers that the apparent affinities we provide are merely estimates. In this context we also wish to remind the reviewer that on a qualitative level (relative affinities), our data from various methods are internally consistent.

15.6. Please either provide the exact equation used to calculate Kds (with the description of parameters) or a reference (less preferred) in the corresponding part of the MMs.

This equation has been added to the Materials section.

16. Lines 175-177. "This positioning interaction pattern were both virtually identical in previous crystal structures of ExoS derived peptides in complex with 14-3-3 proteins (Fig. 3).10,12" The statement is unclear, please rephrase.

An "and" had been omitted; we corrected this.

17. I wonder why the interfaces with ExoS and -T at the "novel" binding site (with a8 and a9 helices of 14-3-3) are not compared in the paper – the interaction is only analyzed in detail using ExoS as an example.

We did mention ExoT explicitly and throughout this paragraph, and the new Supplementary Fig. 5 should contribute to clarity regarding the fact that the binding modes are overall very similar between ExoS and ExoT. However, and as the reviewer points out in his comment 21 below, our structure of the ExoT:14-3-3 complex is not of sufficient quality to provide a basis for a detailed analysis of the interaction site without over-interpretation of existing data. We have highlighted the 14-3-3 interacting residues at the new binding site in the sequence alignment (Supplementary Fig. 6), and included a brief discussion in the main text.

17.1. How conserved are the local contacts in these two cases (overall folds of the ExoS and ExoT complexes with 14-3-3b look similar from Fig. 1)?

Covered by previous comment.

17.2. Is it possible to confirm the universal principle of binding in this outside binding site for all 1433 dependent toxins, based on the differences and similarities of their sequences in the region involved in the novel binding? This comparative analysis seems important for the logics of the paper and for the putting forward the mechanism of toxin activation and 14-3-3 dependence.

Covered by previous comment.

17.3. Along these lines, 14-3-3 independent toxin Vis (lacking LDLA boxes 1 and 2) still has considerable homology with the ExoS and ExoT toxins (suppl. Fig. 5) and therefore is expected to also interact at the “novel” binding site on C-terminal helices of 14-3-3. Is this the case? Otherwise, it would be informative to analyze local differences of Vis (pdb 4xzj) and ExoS/T to rationalize the inability of Vis to bind 14-3-3.

This is not the case (see new Supplementary Fig. 6): The homology between Vis and the 14-3-3 dependent toxins does not extend to this surface.

18. As said before, the hypothesis about the chaperoning function of 14-3-3s on ExoS, which is claimed to be aggregation prone, is rather weak as the experimental verification of this important statement is superficial. The main evidence is DLS data showing the modest stabilizing effect (by only ~4C) of 14-3-3 on “Tagg” of ExoS. This part has technical problems and is incomplete.

18.1. Technically, how was Tagg calculated? This is a rarely used parameter to characterize aggregation (in contrast to denaturation) because it is often difficult to identify the region of aggregation curve saturation, which would be necessary to derive the half-transition parameter, because the last part of the aggregation curves are typically recorded either in already turbid solutions with extremely high fluctuations of the signal or in solutions with no turbidity, but thus in the absence of saturation whatsoever. This seems exactly the case for the curves recorded in the presence of various 14-3-3 concentrations, for example, the highest ratio (2.0) curve shows no saturation and therefore no T0.5 parameter can be correctly determined by sigmoidal curve fitting, as it is stated in MM section. Much better and robust parameter would be temperature at which aggregation starts to occur (possible to determine in the absence of saturation). This could be readily determined by extrapolating the rising part of the thermal curve backwards to the X axis (will be something like 45C in the case of ExoS alone from Fig. 4i).

We agree with the reviewer; this is an excellent suggestion. In the new version of the manuscript, the aggregation onset temperatures are given instead (inset Figure 4i and the new Supplementary Table 3).

18.2. Next problem is that ExoS starts to aggregate at only 45C, which means it is a rather stable protein, at least under physiological conditions, not necessarily aggregation-prone.

Regardless of whether or not one would characterize ExoS as an aggregation prone protein, our data show that ExoS aggregation is alleviated by the presence of 14-3-3 in a dose dependent manner.

18.3. Another issue concerns the fact that the DLS data for only ExoS with two LDLA boxes are presented and therefore no conclusion can be drawn for the ExoS ART domain (lacking LDLA boxes). To address this, the same analysis should be done for the individual ART domain (in the absence or presence of various 14-3-3 concentrations), which would tell whether 14-3-3 affects the aggregation behaviour of ExoS in the absence of the binding at the amphipathic groove(s).

We have carried out this experiment (as already mentioned in the response to the initial comment above). The result (new Supplementary Fig. S14) shows that free 14-3-3 can indeed stabilize the ExoS construct lacking the C-terminal extension during heat induced aggregation.

18.4. Next important thing that is currently lacking is the analysis of the oligomeric status of native (non-aggregated) ExoS (both, containing and lacking LDLA boxes) depending on protein concentration, which would support the categorical conclusion about the aggregation propensity of ExoSs. The doubt is whether in in vivo situation these toxins indeed suffer from this speculative aggregation propensity; overexpression of the recombinant proteins often leads to their increased aggregation, not necessarily because of their intrinsic propensity to aggregate.

Our conclusions are not “categorical”; they are based on six years of experience working with various exotoxins, and experimental results presented in the manuscript. Our model for ExoS activation rests upon observations made in in vivo assays (bacteria and yeast) showing that ExoS containing at least LDLA-box 1 can be activated by 14-3-3; ExoS lacking the C-terminal extension is active when co-expressed with 14-3-3 but inactive and unable to be activated when expressed alone. We have not made any observation suggesting that presence of the C-terminal segment could prevent aggregation caused by protein overexpression per se (i.e., expression in the absence of 14-3-3).

18.5. Ideally, if the aggregation propensity of ExoS/T is confirmed under more natural conditions, the nature of this propensity is good to address to distinguish between self-association of the native-like protein particles (due to their enhanced surface hydrophobicity) or aggregation accompanying their denaturation.

Point taken; this may be useful in future studies concerning PPI inhibitors of ExoS activation.

18.6. Is it possible to speculate that 14-3-3-independent ART-containing toxins (lacking LDLA boxes) are in fact dependent on 14-3-3 chaperone-like activity?

We do not believe there is published evidence for such a conclusion. For example, Rod Merrill has studied a large number of ADP-ribosylating toxins; most of them have kinetic properties in the vicinity of 14-3-3 activated ExoS/ExoT, although 14-3-3 activation was not tested in experiment. Our results, and the model based on them, suggests the following common denominators for 14-3-3 activated proteins: (i) Insignificant catalytic activity on verified targets in absence of 14-3-3. (ii) At least one LDLA-motif on the C-terminal side of the catalytic core. The latter is now well defined thanks to our crystal structures. (iii) Significant exposed hydrophobicity on the central β -sheet (strands 4, 7 and 8), which given our structures is now well enough defined for comparison using a structure based sequence alignment, as in our Supplementary Figure 6.

18.7. It is strange that the hypothesis about the chaperoning activity of 14-3-3s comes “out of the blue”, without being even mentioned in introduction or earlier sections of the study. In fact, chaperone-like activity is a relatively poorly characterized, moonlighting activity of 14-3-3 proteins, different in many aspects from the main phosphopeptide-binding activity (ref 5 of the paper). Despite the current paper uses the terms and hypothesizes that this activity of 14-3-3 with respect to ExoS/T takes place, it does not properly introduce the chaperone function of 14-3-3s in the paper. Please, discuss the problem and the role for hydrophobic clusters in aggregation and chaperone activity more, referring to the research you already cite (ref. 5).

Our hypothesis is not about “the chaperoning activity of 14-3-3s”. Instead our hypothesis is about the 14-3-3 mediated activation of ExoS. It is based on our experimental evidence. Nevertheless, we do

not wish to take credit for the idea that 14-3-3s can act as chaperones and have clarified this by rephrasing the introduction and tying reference 5 into a chaperoning context.

19. Discussion (ca 250 words). I don't know the purpose of these parentheses, but the actual discussion is >700 words.

This has been deleted.

20. Discussion of the "novel binding site".

20.1. Line 120. It is stated that the novel site can be sub-divided into two parts. It is not immediately clear from the Figure 2 what authors mean, please consider marking those two sub-parts in the figure.

This should be clear from the new version of the figure.

20.2. I do not agree that the binding site is really novel and that it only slightly overlaps with the sites found, for example, in 14-3-3 complexes with HSPB6 (pdb 5ltw). Supplementary fig. s12c already contradicts this statement, showing a nice overlap of the binding surfaces, comparably with that in the case of trehalase (pdb 5n6n) – fig. s12a.

We believe that readers will agree on the novelty of the binding site for exotoxins presented by our crystal structures.

The reviewer addresses a different topic, namely, the localization of the surface on 14-3-3 that mediates the interactions with the respective proteins. Naturally, the extent of this surface is impossible to appreciate based on 2D images. We have calculated the common surfaces shared between ExoS and HSPB6: ExoS interacts on an area of 1558 Å²; HSPB6 on an area of 1063 Å²; 479 Å² of these are shared. With these figures, we believe our interpretation is valid and we leave it up to readers to draw their own conclusions based on the pdb entries and literature on HSPB6, which we have cited.

20.3. I especially disagree with the novelty of the idea that "In a development of pharmacological interventions involving the novel exotoxin binding site" and the originality of the statement that "it will be important to address the question of whether other proteins interact at this site". In fact, it has been recently discussed that i) interaction of partners outside the amphipathic grooves is quite often, ii) that a8-a9 helices often take part in such associations and iii) that, considering the lower affinity and higher variability of binding using a8-a9 helices of 14-3-3 (compared to the binding at the amphipathic grooves), drug targeting of such secondary interactions is promising (PMID: 29180038). Please, cite this preceding work which falls exactly in the topic and discussion of your paper. In light of these facts, targeting of the outer surface of a8-a9 helices of 14-3-3 to prevent toxin binding may also be challenging because other (vital) interactions besides those with toxins may be inhibited, therefore the statement "This is especially important as compounds binding to an exotoxin specific site would circumvent the off-target effects expected of inhibitors that bind in the amphipathic groove of 14-3-3 proteins. 29". (lines 307-309) is rather applicable to both interactions, at the amphipathic groove and outer surface of 14-3-3.

The statement in question is factually correct as it stands. We do not claim novelty for the general idea of pharmacological inhibition of PPI outside the phosphopeptide binding site. (We do provide

references to original work for this; the reviewer suggests reference to a review article that deals with the recognition of phosphorylated proteins by 14-3-3.) We may rightfully claim novelty for the idea of pharmacological intervention involving the novel exotoxin binding site, since we have discovered the site in the present work.

We agree with the reviewer's view that "targeting of the outer surface of $\alpha 8$ - $\alpha 9$ helices of 14-3-3 to prevent toxin binding may also be challenging". But we maintain our position on the attractiveness of the site for pharmaceutical intervention: Only a handful of interactions have been defined around that site to date, and unless there will be a landslide of discoveries of PPI at that site, an inhibitor targeting the site will certainly be more selective than one targeting the phosphopeptide binding site. The same general conclusion has been reached by others (Stevens et al 2017; our previous reference 29).

21. Crystallographic part has some inaccuracies and minor problems that need to be addressed.

21.1. It is a complete mystery what was the purpose of obtaining the co-crystal structures of ART domains and the STO1101 compound as it is only stated (lines 100-104) that the structures were obtained and this is not discussed in the paper at all.

STO1101 is a moderately potent ExoS/T inhibitor that has been characterized by us before (our reference 19). In the present study, it was instrumental in determining the structure of ExoT and also the initial structures of ExoS. In light of our ExoS structures in complex with carba-NAD, and in light of the focus of this work on the opportunity to develop inhibitors that bind at a different site, we do not believe it is warranted or meaningful to include an in-depth discussion of STO1101. We have now included a remark on this in the legend to Supplementary Figure 2.

21.2. Considering the crystal structure with STO1101, it has too low resolution (3.2 or 3.8 Å only) to confidently place the inhibitor and reveal the contacts (as e.g. shown in Fig S2c and d). Please show the omit electron density map for the bound inhibitor to support the claims.

This information was added to Supplementary Figure 2. The experimental electron density supports positioning of the ligand STO1101 (see Supplementary Fig. S2c) and is also fully supported by our structure-activity relationship studies of this compound series (PMIDs 26850638 and 29207339).

21.3. It is not clear why the heterotrimeric complex (83 kDa) was first solved using datasets from Bessy (but not refined – first column of the Table I is half empty) and then – using data collected in Diamond, and both datasets are presented in Table I. The different unit cell parameters (primarily beta-angle – 112 and 126 degrees) could be noticed, doubting whether the two crystal forms were completely isomorphous and the same. Please, either remove the first column (and its description in MM) or refine and show the full column in Table I.

As explained in the Methods section, we initially solved the heterotrimeric complex using Vis toxin as molecular replacement search model for ExoS. The following structures were solved by using this initial ExoS structure as search model. Thus, in our view, this is experimental data that should be shown. We solved this problem by placing the unrefined data from the initial complex ("half-empty column") into a separate table in the Supplementary Information document (new Supplementary Table I).

21.4. The last column of Table I indicates that this structure is considered refined and even about to be (or already) deposited with the PDB, despite the text states that the structure had poor resolution and the refinement was abandoned. Of course at this low resolution the refinement is challenging, but the Rfree of 40 cannot be considered good. Please, either try refining the structure more or remove the second part of the last column in Table I. Otherwise the structure at 3.8Å and Rfree of 40% will not be informative and useful.

Low resolution data of a novel structure in the PDB may very well be helpful for others, especially since the experimental evidence is accessible; over-interpretation of experimental data, on the other hand, must be avoided. We use this structure of ExoT to make two important points in this manuscript: (i) The structures of the 14-3-3 complexes with ExoS and ExoT are overall very similar. (ii) The positioning of the N-terminal helices in ExoT looks different than that modelled based on C3 toxin in a previous study. Our experimental density of ExoT is sufficiently well defined to draw these two conclusions.

21.5. The scheme on Fig. 1a is misleading because from it it follows that truncation of several residues from the C-terminus of 14-3-3beta results in switching of the stoichiometry of its complexes with ExoS (from heterotetramer to heterotrimer).

We understand the concern. We have replaced the labels “heterotrimer” and “heterotetramer” with the pdb entries that resulted from the respective expression constructs. The remainder should be clear from the main text and method section.

21.6. It remained unclear from the paper, what was the determining factor of the formation of either trimeric and tetrameric complexes with 14-3-3. Was it just a spontaneous event and these complexes formed spontaneously and then could be resolved by heparin chromatography? Please, make this point more clear.

Our observations (i.e., two types of chromatography) indicate that heterotrimers, heterotetramers, and larger molecular weight complexes all coexist and can be isolated separately. Upon scrutiny of the text we believe that this circumstance should be clear to readers as is (first paragraph of the Results section, and the accompanying Supplementary Figure 3).

Supplement.

Line 61. Reference to Fig. S2 should be to S3. Otherwise, change S2 and S3.

Line 67. Reference to Fig. S1, should be to S3.

Fig. S1b. delta isoform is indicated, supposed to be gamma instead.

Fig. S3. Panels a-d. OY axes labeled inconsistently (Abs280 and UV), why so?

These mistakes have been corrected.

What could explain different toxicity of ART domains of ExoS and ExoT for E.coli cells, which resulted in the ability of authors to obtain only ExoTwt, whereas ExoS needed mutations in the active site to overcome toxicity?

We have not investigated this and can only speculate. Since ExoS and ExoT have different targets in the host cell, a reasonable explanation might be that ExoS ADP-ribosylates an E.coli protein thereby disrupting an essential function, whereas ExoT does not.

Fig. S3e. Could some different tint of purple/pink/violet used to color 14-3-3 dimer in the tetrameric symmetry mate? Now it is difficult to understand where there are 14-3-3 dimer and ART domain monomers.

This panel has been re-drawn for clarity.

Legend to Fig. S5. Lines 125-127. "14-3-3 dependent".. the word "toxins" seems to be missing

This has been corrected.

Line 137. "or LDLA-boxes 1 and 1" - should be 1 and 2?

This has been corrected.

Fig. S8. Please comment on the presence of multiple bands on the ExoS lane? Does it reflect poor purity of the sample?

We suspect that these are co-purifying proteins (e.g., E.coli chaperones) and/or ExoS degradation products containing automodification sites, which get modified when ExoS is activated by 14-3-3 addition during the assay.

Fig. S9. Why there is no protein present in the first band (ExoS)?

We frequently observe exotoxins "disappear" especially at low concentration and after prolonged incubation, presumably due to aggregation and adherence to laboratory plastics.

Why is the 14-3-3 band to the right of the standards fluorescent (only in fig s9 out of s8-s11)? Does it also serve as a substrate for ADP-rib reaction along with K-Ras (only under these specific conditions)? Please comment on this.

Careful measurement of the migration patterns in these SDS-PAGE gels suggests that this is the same protein in all gels shown (upper band in the doublet around the 28 kDa size marker, which is visible only by in-gel fluorescence, not by Coomassie staining, and which is present at varying amounts). We have been unable to identify this protein by mass spectrometry analysis. We believe it may represent either a contaminant or (given the possibility of a change in migration pattern upon ADP-ribosylation) 14-3-3 itself. We have added this information to the figure legend of the new Supplementary Figure 10 and marked the respective bands with asterisks.

Fig. S12. Legend. A bunch of mistakes "14-4-4". Similar attempt to overlay existing complexes and to identify binding regions outside the amphipathic grooves has already been done recently in (PMID: 29180038), please cite this work.

These mistakes have been corrected.

We appreciate the suggestion to include a reference; we respectfully note that we treat this suggestion at our discretion and only within the current context, which is the overlap of the ExoS binding site with that of other 14-3-3 partner proteins.

Fig. S13. Please indicate the position of the A-loop, which is discussed in the legend on structural pictures.

The ARTT-loop has been labelled in all panels.

Nikolai Sluchanko

Reviewer #3 (Remarks to the Author):

Summary:

This study advances understanding of how ExoS and ExoT ADP ribosyltransferase activity is modulated by 14-3-3 proteins. The authors provided cocrystals of adequate resolution to support their findings, and then tested key interactions of residues by mutagenesis or truncations. They identified a new region of interaction between ExoS/T and 14-3-3b in the second LDLA box at the C-terminus in one co-crystal structure, show small reduction in affinity to 14-3-3b, and speculate on implications from this. They are able to construct a 14-3-3b-independent ExoS by making a fusion with Vis toxin. Given the retained affinity for 14-3-3b to ExoS lacking both LDLA boxes, they investigate whether enzymatic activity is still possible when both proteins are co-expressed. The overall findings advance the field in two ways: 1. they have provided a complete ExoS ART domain structure while also confirming partial structures of fragments from previous publications, and 2. They show that the mechanism of activation by 14-3-3b appears to be though alleviating aggregation. Both are sufficiently novel contributions, and of importance to the bacterial protein toxin field and *P. aeruginosa* pathogenesis. The proposed mechanism of 14-3-3 as a chaperone for ExoS rather than a inducing a conformational change to the active site may be of particular interest to a wider audience.

Specific comments:

1. The activity of the ExoS-Vis fusion protein is interesting and unexpected, even with its reduced activity. Is the Vis C-terminus fusion sufficient to alleviate aggregation, too?

Owing to the low yield of recombinant fusion protein we cannot derive a definite answer, because the assay format demands protein of high purity. The fusion protein does have a tendency to deteriorate (we carried out all measurements on freshly purified protein) but this may easily be due to the artificial construction of the C-terminus.

In the meantime, we have addressed the contribution of LDLA-motif containing C-terminus to the anti-aggregating effect by new experiments (see reviewer 2, initial comment and comment 18.3 above).

2. One of the key findings shows that ExoS 233-419 (lacking LDLA motifs) is still active when co-expressed. It is not immediately clear how the co-expressed proteins could be obtained for this experiment in the text given that they are toxic to *E. coli*; the methods indicate a shorter induction with increased IPTG, which presumably got around this problem. If that is correct, a short statement preceding line 256 in the text could avoid confusion.

That is correct. We have added a clarifying statement in the text, as suggested.

3. Related to point 2, this particular result (catalytically active ExoS 233-419 when co-expressed with 14-3-3b) seems unexpected given ExoS 233-419 is not toxic in yeast in Figure 3e. Perhaps the authors can hypothesize on why ExoS 233-419 is not toxic in yeast, or has no partial effect, even though it might be catalytically active.

These two test systems (co-expression in bacteria and overexpression in yeast) are sufficiently different to make speculation about the reason for the lack of ExoS 233-419 yeast toxicity very difficult. The short construct may bind to the yeast 14-3-3 proteins with lower affinity than to the human orthologs. There may be a spatial aspect, such that toxin production is in a subcellular region that does not allow immediate exposure to free yeast 14-3-3 protein. We also know that there is a temporal aspect, because full length ExoS ART domain production kills yeast cells efficiently but only after several hours of induction (our results not shown in the manuscript). Finally, co-expression from the same plasmid might be required to achieve toxicity. Thus we believe it is appropriate to meet this referee's suggestion by pointing out that the two experimental systems are sufficiently different to preclude a straightforward conclusion. This has been added in the discussion section.

4. A critical component of the authors' model depicting 14-3-3b as a chaperone depends on a single dynamic light scattering experiment and is only otherwise supported by their observation of inclusion bodies. Perhaps this finding could be strengthened if a negative control, such as an irrelevant protein, is also shown in addition to the titration of 14-3-3b.

We have conducted this experiment and added the results along with our new DLS data (Supplementary Fig. 14 and Supplementary Table III).

5. The authors are a bit sparse on citations of previous work in their introduction. For example, they do not cite discovery of the *P. aeruginosa* T3SS nor any of the work identifying targets of ExoS and ExoT. However the writing is of good quality and will be easy to follow by those in the field.

We have now expanded the introduction according to both reviewers' suggestions, and added eleven new references to key work in the field.

Sincerely,

Herwig Schüler

Reviewers' comments:

Reviewer #2 (Remarks to the Author):

The authors did a great job on improving their manuscript and I appreciate that they took benefit from my recommendations. At the same time, the rebuttal letter and the revised MS revealed several remaining issues and I'm afraid that an additional iteration is required.

1. Important point (related to the previous P.15.5).

Although the data on the binding affinity of the ExoS constructs to 14-3-3 indeed seem internally consistent and qualitatively say that the LDLA boxes slightly increase the affinity and in combination can even lead to some avidity effects making the toxin with both C-terminal LDLA boxes the tightest 14-3-3 binder, I insist that a control would be informative to get an idea of how tight the novel 14-3-3/ExoS binding really is. If the authors opt out the suggested use of additional methods (commonly used in other works on 14-3-3) to support their submicromolar affinities, please use an alternative approach, that is, provide an estimation of the apparent affinity using your technique (fluorescence polarization), but on an already characterized 14-3-3 partner protein or peptide. This is especially important in the light of the facts indicated in the rebuttal that just 100 molecules of the toxin may be enough to cause its toxic action. If the number is that low, the "sequestering model" seems less likely (still, is worthy of some discussion in the MS). On one side, to outcompete hundreds of normal cellular partners of 14-3-3, the toxin, if present in such minor quantities, needs to have an extremely high affinity to 14-3-3 to ensure formation of stable complexes mediating its action. On the other side, such a low intracellular quantity of toxin would not exhibit aggregation and would not probably require 14-3-3 for chaperoning. Thus, the direct comparison of the ExoS/T binding to 14-3-3 with the binding of any other previously characterized partner is a rather important logical point.

2. Important point (related to the previous P.17.3. and P.18.6).

Still, is it possible to clearly rationalize the 14-3-3-independence of Vis toxin (and others) from the point differences in the region that in 14-3-3-dependent toxins binds at the "novel site"? (e.g., some key contacts between b4/b7/b8 of ExoS and a8/a9 in 14-3-3 are different in Vis making its binding improbable)

3. Important point (related to the previous P.18.2. and 18.4.).

The point was that if a protein starts to aggregate in vitro only at temperatures higher than 45C, can it be called "aggregation-prone" (was done so in the legend of Fig. 4j)? Not necessarily. A brief mentioning of the fact that the protein was indeed always tending to aggregate would be useful. More importantly, does ExoS aggregation occur in (near) natural conditions, given its low copy number (as stated by the authors in the rebuttal letter) and the lower temperature in the host?

Minor points.

p.3 bottom. Reference 17 is slightly outdated (2011), that time the only known crystal structure for a 14-3-3/phosphoprotein complex was that with AANAT (2001), whereas in the past 7 years at least three other structures of 14-3-3 complexes appeared (with Hd3a, HSPB6, NTH1). A more recent overview on 14-3-3/phosphoprotein associations is recommended for referring a reader to. Fig. 4b and c. epsilonNAD (OY, b) and eNAD (OX, c) inconsistent designations are used. Fig. 4i and S14. Particle size diameter (nm) -> Particle diameter (nm). Size is a redundant word here.

p.15, tenth row from bottom. NAD+ glycohydrolaseS activity -> NAD+ glycohydrolase activity.

Methods. Enzymatic analyses. 14.3.3b -> 14-3-3b (two instances).

Fig. S4. Please indicate positions of the proteins by arrows. Unfortunately, the provided picture does not show any differences in the ratio of the protein bands for the heterotrimeric and heterotetrameric complexes, i.e., the pattern of overloaded lanes B10 (trimer) and C4 (tetramer) is almost identical.

The legend to Fig. S4. "Protein complex was isolated from E.coli lysates by IMAC, and subjected to SEC, as described in the methods section. The figure shows the purity of the material that was loaded onto a heparin column". The authors describe the purification procedure as composed of only IMAC+SEC in the first sentence, but loading onto heparin column in the second sentence and after that. The heparin column was between IMAC and SEC, right?

Again, Fig. S15. 14-4-4... (seven instances!). Please provide also the numbers describing the overlap in the contacting area between the structures in the region outside the amphipathic groove that are presented in the rebuttal letter. This will indeed make the comparison more concrete.

As in the revised version of the MS the main messages of the authors became much clearer, I would like to attract their attention to the following terminology issue. The "reconstituted complex" in the last part of the results and in Fig. 4j, i.e., that formed from mixing of separately produced

14-3-3 and ExoS-ART, is a misleading term because the main idea, as could be inferred from the proposed model, is that the discussed complex was in fact not formed (after ExoS-ART is already aggregated in the absence of 14-3-3) unless the LDLA boxes were present in the construct or it was co-expressed with 14-3-3. This has a direct relation to the differences in the so-called "holdase" and "disaggregase" chaperone-like activity, where the first one prevents aggregation of a substrate protein (being unable to resurrect it from aggregates) and the second one efficiently works with the already aggregated protein substrate (usually in an ATP-dependent manner). In other words, the term "reconstitution" (as an attempt) is OK, but the formation of the complex by reconstitution apparently failed and therefore the "reconstituted complex" in this context is an absurd concept. Otherwise, there should be no difference in whether the complex is co-expressed or reconstituted as long as it is formed and 14-3-3 can affect the toxin's activity.

--
NS

Reviewers' comments:

Reviewer #2 (Remarks to the Author):

The authors did a great job on improving their manuscript and I appreciate that they took benefit from my recommendations. At the same time, the rebuttal letter and the revised MS revealed several remaining issues and I'm afraid that an additional iteration is required. Thank you for a second scrutiny of our manuscript, which has certainly been instrumental for further improvements.

1. Important point (related to the previous P.15.5).

Although the data on the binding affinity of the ExoS constructs to 14-3-3 indeed seem internally consistent and qualitatively say that the LDLA boxes slightly increase the affinity and in combination can even lead to some avidity effects making the toxin with both C-terminal LDLA boxes the tightest 14-3-3 binder, I insist that a control would be informative to get an idea of how tight the novel 14-3-3/ExoS binding really is. If the authors opt out the suggested use of additional methods (commonly used in other works on 14-3-3) to support their submicromolar affinities, please use an alternative approach, that is, provide an estimation of the apparent affinity using your technique (fluorescence polarization), but on an already characterized 14-3-3 partner protein or peptide.

We have not been clear enough in pointing to the fact that we did determine 14-3-3:ExoS affinity employing an alternative method: the 14-3-3 dose dependent activation of ExoS activity. The apparent K_d values are in reasonable agreement with our fluorescence anisotropy measurements (~20 nM for the longest construct and 14-3-3beta). We have now incorporated these results in a new panel adjacent to the anisotropy panels in Figure 3. The above concerns the longest construct – ART domain with full C-terminus – the physiologically most relevant one that has also been subject of the present discussion and for which enzymatic activity can be measured. Concerning the LDLA-box 1 and -2 containing tail peptide there is independent information indicating that our affinity measurements are in the correct range: A peptide “QGLLDALDLAS” was used by Cromm et al., Grossmann (PMID: 27596722) who determined a K_d value of 0.81 μM for the zeta isoform of 14-3-3, also using fluorescence anisotropy. We consider this result consistent with our own data, which indicated somewhat higher affinity, still in the sub-micromolar range, for the segment containing both LDLA motifs. We have added this information and a reference to the above paper in the legend to Supplementary Figure 8.

This is especially important in the light of the facts indicated in the rebuttal that just 100 molecules of the toxin may be enough to cause its toxic action. If the number is that low, the “sequestering model” seems less likely (still, is worthy of some discussion in the MS).

The “sequestering model” (i.e., ExoS causes toxicity via preventing 14-3-3 proteins from carrying out other vital functions) has been formulated by this reviewer in his original criticism. We respectfully note that we do not subscribe to such a model and we consider our previous explanation as sufficient.

On one side, to outcompete hundreds of normal cellular partners of 14-3-3, the toxin, if present in such minor quantities, needs to have an extremely high affinity to 14-3-3 to ensure formation

of stable complexes mediating its action. On the other side, such a low intracellular quantity of toxin would not exhibit aggregation and would not probably require 14-3-3 for chaperoning. Thus, the direct comparison of the ExoS/T binding to 14-3-3 with the binding of any other previously characterized partner is a rather important logical point.

The detailed balance principle dictates that complex formation - enough for ExoS activation- will occur even among multiple components with similar affinities. (Incidentally, considerations about enzymatic activity are equally important and in contrast to many NAD^+ consuming enzymes the exotoxins have a K_M value well below the intracellular $[\text{NAD}^+]$, Table 2.) We do not postulate a need for stable complexes during cellular invasion; however, our data certainly support the formation of stable complexes: As described in the paper, heterotrimers, heterotetramers, and larger complexes were isolated by a hexahistidine tag only on ExoS/T, and were then stable over three further consecutive chromatography steps (SEC – IEX – SEC) at sub-micromolar protein concentration. Again, the main point of our ExoS aggregation-recue model is to incorporate our experimental findings in a new context, most importantly, 14-3-3 proteins do not activate ExoS by inducing a conformational change that forms a competent active site. The underlying mechanistic details – including aggregation kinetics addressed by the reviewer – will need to be studied under suitable experimental settings in the future.

2. Important point (related to the previous P.17.3. and P.18.6).

Still, is it possible to clearly rationalize the 14-3-3-independence of Vis toxin (and others) from the point differences in the region that in 14-3-3-dependent toxins binds at the “novel site”? (e.g., some key contacts between b4/b7/b8 of ExoS and a8/a9 in 14-3-3 are different in Vis making its binding improbable)

We agree with the reviewer that it appears likely (based on the structure based sequence alignment in our Supplementary Figure 6) that Vis cannot interact at this site. However, Vis and others have already been thoroughly analyzed as 14-3-3 independent toxins. We do not question the published research; neither do we believe that the 14-3-3 independence needs to be rationalized, in particular in the absence of further bioinformatics and experimental data.

3. Important point (related to the previous P.18.2. and 18.4.).

The point was that if a protein starts to aggregate in vitro only at temperatures higher than 45C, can it be called “aggregation-prone” (was done so in the legend of Fig. 4j)? Not necessarily. A brief mentioning of the fact that the protein was indeed always tending to aggregate would be useful. More importantly, does ExoS aggregation occur in (near) natural conditions, given its low copy number (as stated by the authors in the rebuttal letter) and the lower temperature in the host?

This label (“aggregation prone”) is used strictly in the context of our model; it is how we envision exotoxin activation to occur in general terms; we believe that readers will be able to make this distinction, and ask you to again consider our statement in response to point 1 in this context. Under laboratory conditions, many recombinant proteins have a tendency to aggregate and the exotoxins are no notable exception that needs to be described further, in our view.

Minor points.

p.3 bottom. Reference 17 is slightly outdated (2011), that time the only known crystal structure for a 14-3-3/phosphoprotein complex was that with AANAT (2001), whereas in the past 7 years at least three other structures of 14-3-3 complexes appeared (with Hd3a, HSPB6, NTH1). A more recent overview on 14-3-3/phosphoprotein associations is recommended for referring a reader to.

Reference 17 is cited for the review of client protein derived peptide complex structures. For the protein complexes we cite original work (refs 36-39) and we have added a reference to the reviewer's own recent review in that position in the text (new ref. 40).

Fig. 4b and c. epsilonNAD (OY, b) and eNAD (OX, c) inconsistent designations are used.

Fig. 4i and S14. Particle size diameter (nm) -> Particle diameter (nm). Size is a redundant word here.

p.15, tenth row from bottom. NAD+ glycohydrolaseS activity -> NAD+ glycohydrolase activity.

Methods. Enzymatic analyses. 14.3.3b -> 14-3-3b (two instances).

These errors have been corrected.

Fig. S4. Please indicate positions of the proteins by arrows.

Arrows have been added.

Unfortunately, the provided picture does not show any differences in the ratio of the protein bands for the heterotrimeric and heterotetrameric complexes, i.e., the pattern of overloaded lanes B10 (trimer) and C4 (tetramer) is almost identical.

Differences in ratios can be appreciated from the early respectively late fractions of the two major peaks (B6 and C10).

The legend to Fig. S4. "Protein complex was isolated from E.coli lysates by IMAC, and subjected to SEC, as described in the methods section. The figure shows the purity of the material that was loaded onto a heparin column". The authors describe the purification procedure as composed of only IMAC+SEC in the first sentence, but loading onto heparin column in the second sentence and after that. The heparin column was between IMAC and SEC, right?

The procedure was IMAC-SEC-heparin and then a second RALS coupled SEC step of each heparin peak. (The last SEC-RALS step was initially carried out for analytical purposes to understand the nature of the peaks eluting at different salt concentrations from heparin, and was later used to isolate the separate species of various size.) Thus the legend states the procedure correctly: We show the SEC pool that went on heparin, and the heparin fractions that then went on to SEC-RALS for molecular size determination.

Again, Fig. S15. 14-4-4... (seven instances!).

This has been corrected. Also, the corresponding author has been persuaded to enroll in a course in text processing software.

Please provide also the numbers describing the overlap in the contacting area between the structures in the region outside the amphipathic groove that are presented in the rebuttal letter. This will indeed make the comparison more concrete.

We have added this information to the figure legend.

As in the revised version of the MS the main messages of the authors became much clearer, I

would like to attract their attention to the following terminology issue. The “reconstituted complex” in the last part of the results and in Fig. 4j, i.e., that formed from mixing of separately produced 14-3-3 and ExoS-ART, is a misleading term because the main idea, as could be inferred from the proposed model, is that the discussed complex was in fact not formed (after ExoS-ART is already aggregated in the absence of 14-3-3) unless the LDLA boxes were present in the construct or it was co-expressed with 14-3-3. This has a direct relation to the differences in the so-called “holdase” and “disaggregase” chaperone-like activity, where the first one prevents aggregation of a substrate protein (being unable to resurrect it from aggregates) and the second one efficiently works with the already aggregated protein substrate (usually in an ATP-dependent manner). In other words, the term “reconstitution” (as an attempt) is OK, but the formation of the complex by reconstitution apparently failed and therefore the “reconstituted complex” in this context is an absurd concept. Otherwise, there should be no difference in whether the complex is co-expressed or reconstituted as long as it is formed and 14-3-3 can affect the toxin’s activity.

We have exchanged the term “reconstituted complex”. The sentence now reads “...whereas ExoS233-419 when reconstituted with 14-3-3beta was inactive.” We have made similar changes in other parts of the manuscript.

--

NS